

# Implementation of methane cycling for deep time, global warming simulations with the DCESS Earth System Model (Version 1.2)

Gary Shaffer[1,2], Esteban Fernández Villanueva[3], Roberto Rondanelli[3,4], Jens Olaf Pepke Pedersen[5], Steffen Malskær Olsen[6], Matthew Huber[7,8]

[1]GAIA-Antarctica, Universidad de Magallanes, Punta Arenas, Chile
[2]Niels Bohr Institute, University of Copenhagen, 2100 Copenhagen Ø, Denmark
[3]Department of Geophysics, University of Chile, Santiago, Chile
[4]Center for Climate and Resilience Research, University of Chile, Santiago, Chile
[5]National Space Institute, Technical University of Denmark, 2800 Kongens Lyngby, Denmark
[6]Danish Meteorological Institute, 2100 Copenhagen Ø, Denmark
[7]Earth, Atmospheric and Planetary Sciences, Purdue University, West Lafayette, IN 47907, USA
[8]Institute for the Study of Earth, Oceans, and Space, University of New Hampshire, Durham, NH 03814, USA,

*Correspondence to*: Gary Shaffer (gary.shaffer.chile@gmail.com)

**Abstract.** Geological records reveal a number of ancient, large and rapid negative excursions of carbon-13 isotope. Such excursions can only be explained by massive injections of depleted carbon to the Earth System over a short duration. These injections may have forced strong global warming events, sometimes accompanied by mass extinctions, for example the Triassic-Jurassic and End-Permian extinctions, 201 and 252 million years ago. In many cases evidence points to methane as the dominant form of injected carbon, whether as thermogenic methane, formed by magma intrusions through overlying carbon-rich sediment, or from warming-induced dissociation of methane hydrate, a solid compound of methane and water found in ocean sediments. As a consequence of the ubiquity and importance of methane in major Earth events, Earth System models should include a comprehensive treatment of methane cycling but such a treatment has often been lacking. Here we implement methane cycling in the Danish Center for Earth System Science (DCESS) model, a simplified but well-tested Earth System Model of Intermediate Complexity. We use a generic methane input function that allows variation of input type, size, time scale and ocean-atmosphere partition. To be able to treat such massive inputs more correctly, we extend the model to deal with ocean suboxic/anoxic conditions and with radiative forcing and methane lifetimes appropriate for high atmospheric methane concentrations. With this new model version, we carried out an extensive set of simulations for methane inputs of various sizes, time scales and ocean-atmosphere partitions to probe model behaviour. We find that larger methane inputs over shorter time scales with more methane dissolving in the ocean lead to ever-increasing ocean anoxia with consequences for ocean life and global carbon cycling. Greater methane input directly to the atmosphere leads to more warming and, for example, greater carbon dioxide release from land soils. Analysis of synthetic sediment cores from the simulations provides guidelines for the interpretation of real sediment cores spanning the warming events. With this improved DCESS model version and paleo-reconstructions, we are now better armed to gauge the amounts, types, time scales and locations of methane injections driving specific, observed deep time, global warming events.





## 1 Introduction

Analyses of carbonate and organic carbon samples have revealed, with ever-finer time resolution, a number of large and rapid negative carbon isotope excursions (CIE) since the Cambrian radiation of life about 540 Ma (million years before present). These CIE's had typical $\delta^{13}C$ amplitudes of > 2 ‰ and typical time scales of < 10000 years. Some prominent examples of this have been found at about 56, 120, 183, 201, 252 and 260 Ma (Hesslebo et al., 2002; Jenkyns, 2003; McElwain et al., 2005; Retallack & Jahren, 2008; Ruhl et al., 2011; Shen et al., 2011; Wignall et al., 2009; Zachos et al. 2001). Such large, rapid CIE's can only be explained by large, rapid injections of light carbon (i.e. carbon-13 depleted) to the ocean-atmosphere-land biosphere system, injections that forced global warming events for each of these CIE's. As a starting point the source of this carbon could be some combination of volcanism ($\delta^{13}C$ ~ -7 ‰), terrestrial biosphere reduction ($\delta^{13}C$ ~ -25 ‰), thermogenic methane input (TM; $\delta^{13}C$ ~ -40 ‰) and methane hydrate release (MH; $\delta^{13}C$ ~ -60 ‰), but the incorporation of further constraints often allows us to rule out the first two options as important contributors to most if not all these events.

For example, if we assume an initial carbon pool in the ocean-atmosphere-land biosphere of about 40000 GtC (1 Gigaton of carbon (GtC) is $10^{15}$gC) with a mean $\delta^{13}C$ of 1 ‰ as at present. Mass balance calculations then show that a large CIE of -5‰, like for example that of the end-Permian event (Shen et. al., 2011), could be explained by injections of ~ 44400, 8900, 5300 and 3400 GtC of volcanic carbon, biosphere carbon, TM or MH, respectively. The volcanic carbon explanation seems unlikely due to the required size and speed of the input and an explanation in terms of terrestrial biosphere die off seems unlikely since it would require a biomass at least 4 times greater than the present day biomass of around 2000 GtC (Shaffer et al., 2008). Therefore the most likely explanations for the CIE's and associated global warming events involve TM and/or MH inputs (Berner, 2002; Hesselbo et al., 2000; McElwain et al., 2005; Retallack & Jahren, 2008; Ruhl et al., 2011; Svensen, H. et al., 2004; Shaffer et al., 2016).

TM is produced when magma intrudes into overlying sediments or coal beds containing old organic carbon. More such intrusions would be expected in association with increased volcanism that creates large igneous provinces and such provinces tend to correlate in time with the ancient CIE's (Svensson et al, 2004). MH is a solid compound of methane and water formed for sufficiently high pressure and low temperature and found mainly in ocean sediments. Within a hydrate stability zone, MH is formed when methane release from bacterial remineralization of organic matter exceeds that needed to sustain solubility levels. Warming will lead to MH destabilization and thereby feedback positively on the warming (Dickens et al., 1995). MH dissociation was touted early on as an explanation for some deep-time CIE's in a number of papers (e.g. Hesselbo et al., 2000; Berner, 2002; Kemp et al., 2005).



The above underlines the need to consider input and ocean-atmosphere cycling of methane in modelling of such deep time global warming events, yet in most cases models used to study such events have not included a full prognostic and interactively coupled approach incorporating methane into the climate as well as the biogeochemical cycles and isotope components. In addition to the question of the source of the light carbon inputs that force the events, other important aspects

for understanding the workings of these events include the amounts, timings and locations of these inputs. For example, over what time scales is this carbon injected and how are the inputs partitioned between the ocean and the atmosphere? In order to address these questions, paleodata reconstructions should be compared with event simulations using Earth System Models that consider methane input and cycling. For such a task, important aspects should be addressed like carbon isotope cycling, bacterial oxidation of methane in the ocean, air-sea gas exchange of methane, the dependency of atmospheric methane

lifetimes on methane concentration there and the extreme radiative forcing for high atmospheric concentrations of methane. However, to our knowledge, such an Earth System Model does not exist at present. Here we upgrade the Danish Center for Earth System Science (DCESS) Earth System Model to fill this gap by designing appropriate methane input functions and by implementing ocean-atmosphere-biosphere cycling of methane. Since large, rapid inputs of methane to the ocean and oxidation of this methane there can lead to suboxic or anoxic oceanic conditions, we also upgrade our model to deal with

ocean denitrification, nitrogen fixation and sulphate reduction (Shaffer 1989). Finally we present some test simulations of deep time, global warming events using this extended model to illustrate ways forward for future model-data analyses.

## 2 The DCESS model up to now

In its original configuration, the DCESS Earth System Model (Version 1.0; Shaffer et al., 2008) features modules for the

atmosphere, ocean, ocean sediment, land biosphere and lithosphere (Fig. 1). The model has been designed to simulate global change on time scales of years to millions of years. It has been calibrated and tested against Earth System data and, where necessary, against output from more complex models (Archer et al., 2002; Schmidt and Shindell, 2003). The model consists of one hemisphere, divided by 52° latitude into a low-mid latitude and a high latitude zone. Global reservoirs, transports and fluxes are obtained by doubling the hemispheric values. The atmosphere module considers radiation balance, meridional

transport of heat and water vapor between low-mid latitude and high latitude zones, and heat and gas exchange with the ice-free part of the ocean. Gases considered are carbon dioxide ($CO_2$) and methane ($CH_4$) for all three carbon isotopes, nitrous oxide and oxygen. Methane is oxidized to $CO_2$ on time scales on the order of ten years, increasing as methane concentrations increase (see below). The model also deals with oxygen isotopes in atmospheric water vapor. Sea ice and snow cover are diagnosed from estimated meridional profiles of atmospheric temperature.

The model ocean is 270° wide, extends from the equator to 70° latitude and covers 70.5 % of the Earth surface. Each ocean sector has 55 vertical layers with 100 m resolution. An ocean sediment sector is assigned to each of the ocean layers with widths representing modern day hypsography. The ocean module has prescribed circulation and mixing. Tracers considered



are temperature, salinity, oxygen isotope in water, phosphate, dissolved oxygen, dissolved inorganic carbon for all three carbon isotopes, and alkalinity. Biogenic production of particulate organic matter depends on surface layer phosphate availability but with lower efficiency in the high latitude zone. The calcite to organic carbon rain ratio depends on surface layer temperature.

The semi-analytical, ocean sediment module considers calcium carbonate dissolution and both oxic and anoxic organic matter remineralisation. The sediment is composed of calcite, non-calcite mineral and reactive organic matter. Sediment porosity profiles are related to sediment composition and a bioturbated layer 0.1 m thick is applied. Carbonate and organic carbon burial are calculated from sedimentation velocities at the base of the bioturbated layer. Bioturbation rates and oxic

and anoxic remineralisation rates depend on organic carbon rain rates and dissolved oxygen concentrations. The land biosphere module considers leaves, wood, litter and soil. Net primary production depends on atmospheric $CO_2$ concentration and remineralization rates in the litter and soil depend on mean atmospheric temperatures. Methane production is a small, fixed fraction of the soil remineralization. The lithosphere module considers outgassing, weathering of carbonate and silicate rocks and weathering of rocks with old organic carbon and phosphorus. Weathering rates are related to mean atmospheric

temperatures.

The DCESS model has been applied mainly to long term, future projections of climate and Earth System response to anthropogenic forcing (Shaffer et al., 2009; Shaffer, 2009; Shaffer, 2010). DCESS model results compare favorably with those of the best other Earth System Models of Intermediate Complexity (EMICS) as seen in recent intercomparison studies

of past, present and future carbon cycling and climate (Eby et al., 2013; Joos et al. 2013; Zickfeld et al. 2013). The results from these studies were also included in the Fifth Assessment Report of the Intergovernmental Panel on Climate Change (IPCC, 2013). From the 2013 studies onward, the carbon fertilization parameter of the model terrestrial biosphere module was reduced to 0.37 from the original value of 0.65 (Eby et al., 2013).

The Paleocene-Eocene Thermal Maximum (PETM) warming event was addressed in the most recent model application (Shaffer et al., 2016) whereby the very simplified methane cycling sketched above was implemented and the carbon input was taken to occur over a ten thousand year timescale. That is significantly longer than time scales associated with methane oxidation in the atmosphere or ocean overturning and the system response is rather insensitive to input type and location (ocean or atmosphere). Therefore, for simplicity the carbon input was taken to be in the form of carbon dioxide and was

injected into the atmosphere. To deal with this deep time warming event, several other modifications were made to the basic DCESS model (Shaffer et al., 2016): 1. The solar constant was decreased by 0.5 % from present day, sea level was raised by 100 m while retaining the modern day hypsography, and mean ocean salinity was taken to be 33.8 (the modern value is 34.7), 2. Background albedo was reduced due to less land at subtropical latitudes and more vegetation cover, 3. Initial land biomass was increased to reflect more land at low and mid-latitudes, 4. To account for high latitude cloud forcing, extra



forcing was imposed poleward of the latitude of sea ice extent in the standard model, pre-industrial (PI) steady state, 5. Parameters in the simple Budyko-type formulation for outgoing longwave radiation (Budyko 1969) were adjusted in concert with the other forcing changes to achieve late Paleocene temperatures and climate sensitivity estimates, 6. Ocean calcium and magnesium concentrations were adjusted to late Paleocene values, 7. Model weathering formulation was extended to

5    include the size of the weathering substrate, estimated for the late Paleocene to be less than that for the present, 8. Lithosphere outgassing was increased to conform to late Paleocene estimates, 9. The formulation of the calcite to organic carbon rain ratio was extended to also depend on the calcite saturation state of the ocean surface layer and, 10. The dependence of carbon isotope discrimination during ocean surface layer photosynthesis was extended to depend on concentrations of dissolved carbon dioxide and phosphate there.

## 3  Methane cycling implementation

### 3.1 Atmosphere

For an oxygenated atmosphere the dominant sink for atmospheric methane is oxidation to $CO_2$ by reaction with the hydroxyl (OH) radical. Since this reaction depletes the concentration of these radicals, methane atmospheric lifetime, $\tau$, grows as

15   methane concentrations increase. In the original model (Shaffer et al., 2008), this effect and that of associated chemical reactions in the troposphere and stratosphere was addressed by fitting results from a complex atmospheric chemistry model (Schmidt and Shindell, 2003) to the equation

$$\tau = \tau_{PI}(M+b)/\left[(1-a)M+b\right] \tag{1}$$

where $\tau_{PI}$ is the pre-industrial (PI) lifetime, $a$ and $b$ are fitting constants and $M \equiv (p\mathrm{CH}_4 - p\mathrm{CH}_{4,\mathrm{PI}})/p\mathrm{CH}_{4,\mathrm{PI}}$ where $p\mathrm{CH}_4$ is the atmospheric partial pressure of methane and $p\mathrm{CH}_{4,\mathrm{PI}}$ is its PI value taken to be 720 ppb. The Schmidt and Shindell (2003) target results and the model fit to them used in Shaffer et al., (2008) are shown in Fig. 2 (red dots and blue line). For this fit $\tau_{PI} = 6.9$ years, $a = 0.96$ and $b = 6.6$.

Fig. 2 also shows results from several additional chemistry modeling studies for the dependency of $\tau$ on $p\mathrm{CH}_4$ (blue dots from Lamarque et al. (2006); black dots from Isaksen et al. (2011)). In Lamarque et al, (2006), atmospheric lifetimes decrease for extreme values of $p\mathrm{CH}_4$ due to much enhanced water vapor and OH production (and thereby greater methane oxidation) of a very warm climate. The red line in Fig. 2 shows a fit to all the chemistry modeling results of the figure using

30   Eq. 1, a fit for which $\tau_{PI} = 9.5$ years, $a = 0.78$ and $b = 11$. This new fit now also captures in part enhanced OH production and associated limitation of $\tau$ for very high values of $p\mathrm{CH}_4$ and is more consistent with a multi-model ensemble of models,



reported in Voulgarakis et al. (2013). We have adopted this new fit in our enhanced model whereby the model atmospheric methane sink in moles per year is $\upsilon_a p\mathrm{CH}_4 / \tau$ where $\upsilon_a$ is the atmospheric mole content.

The standard radiative forcing formulation for methane (including spectral overlap with nitrous oxide) can be used for methane concentrations up to about 5 ppm (Myhre et al., 1998), much less than the concentrations we deal with here. However, recent work has provided methane radiative forcing formulations, including nitrous oxide overlap, for methane concentrations up to 100 ppm (Byrne and Goldblatt, 2014a) and even up to 10000 ppm (Byrne and Goldblatt, 2014b). We adopt these results and, for consistency, we also adopt the Byrne and Goldblatt (2014a) expressions for carbon dioxide radiative forcing and for carbon dioxide-nitrous oxide overlap. Figure 3 serves to illustrate these radiative forcings. Note that for very high methane concentrations, radiative forcing levels off and even decreases, an effect caused by enhanced short wave absorption at such concentrations (Byrne and Goldblatt, 2014b). Taken together with the leveling off of methane atmospheric lifetimes at such concentrations (Fig. 2), this limits the role that can be played by methane in enhancing global warming under such extreme conditions. If as an example we consider an atmosphere with partial pressures $p\mathrm{CO}_2$, $p\mathrm{CH}_4$ and $p\mathrm{N}_2\mathrm{O}$ of 1000, 100 and 1 ppm, respectively, this corresponds to radiative forcings relative to pre-industrial (PI) conditions of 7.45, 6.50 and 1.88 W/m$^2$ but with a total forcing overlap of -0.85 W/m$^2$, leading to a total forcing of 14.98 W/m$^2$ (note that the CO$_2$ forcing of 7.45 W/m$^2$ is significantly greater than the value of 6.50 W/m$^2$ that would be calculated using the standard formulation (Myhre et al., 1998; Shaffer et al., 2008)). For a nominal climate sensitivity of 3°C for a $p\mathrm{CO}_2$ doubling (0.81°C/W/m$^2$), such a total forcing would lead to a global mean temperature increase of 12.1°C above the PI mean global temperature of 15°C ; for a 5°C climate sensitivity (Shaffer et al, 2016) the increase would be 20.2°C. Finally note that in this simplified atmosphere module we do not deal with explicit forcings associated with tropospheric ozone and stratospheric water vapor.

## 3.2 Ocean

### 3.2.1 Simplified nitrogen and sulphur cycling

To be able to deal with suboxic and anoxic ocean conditions that would arise for significant oxidation of methane in the ocean interior, the model has been generalized to consider nitrogen and sulphur cycling with the introduction of nitrate (NO$_3$), ammonium (NH$_4$) and hydrogen sulfide (H$_2$S) as additional ocean tracers (Shaffer, 1989; for simplicity, we often omit ion charges here). Furthermore for simplicity we take the river input of nitrogen to be in the form of nitrate and to be equal to the (climate- and weathering substrate-dependent) river input of phosphate multiplied by $r_{NP}$, the N:P Redfield (mole) ratio, taken to be 16. Likewise, organic matter produced, remineralized or buried maintains this ratio of nitrogen to phosphorus. Following Shaffer et al. (2008), we neglect dissolved organic matter such that the rate of export of particulate





organic matter (POM) down out of the surface layer is equal to new production and we assume an exponential law for the vertical distribution of remineralization of POM "nutrient" and "carbon" components, each with a distinct e-folding length, $\lambda_N$ and $\lambda_C$, respectively.

5  Model new production of organic matter in the ocean surface layer, $NP$, has been generalized to depend on the limiting nutrient, either phosphate or nitrate. We take $NP = \min(NPP, NPN)$ where the phosphate- and nitrate-based, new production, $NPP$ and $NPN$, are

$$NPP^{l,h}, NPN^{l,h} = A_0^{l,hni} z_{eu}(1, r_{NP})^{-1}(L_f^{l,h}/sy)(PO_4^{l,h}, NO_3^{l,h})\{\left[PO_4^{l,h}/(PO_4^{l,h}+P_{1/2})\right], \left[NO_3^{l,h}/(NO_3^{l,h}+N_{1/2})\right]\} \qquad (2)$$

whereby $l,h$ refer to low-mid latitude and high latitude ocean zones, $A_0^{l,hni}$ are the ice-free ocean surface areas, $z_{eu}$ is the surface layer depth (100 m), $r_{NP}$ is a Redfield ratio (taken to be 16), $sy$ is the number of seconds per year, $PO_4^{l,h}$ and $NO_3^{l,h}$ are the surface layer, phosphate and nitrate concentrations, and $P_{1/2}, N_{1/2}$ are half saturation constants (1 $\mu$mol/m$^3$, 16 $\mu$mol/m$^3$). $L_f^{l,h}$ are efficiency factors, taken to be 1 and 0.36 for the low-mid and high latitude zones, respectively (Shaffer et

15  al., 2008). This is how the model accounts for light and iron limitation in the high latitude zone.

The surface layer sinks due to new production are $-NP^{l,h}$ for phosphate and $-r_{NP}NP^{l,h}$ for nitrate. Other source/sinks and Redfield ratios are adopted from Shaffer et al. (2008). Since we will deal with water column denitrification, we also consider nitrogen fixation as a surface layer source of nitrate when $NO_3^{l,h}$ levels there drop below $r_{NP}PO_4^{l,h}$ or equivalently when

20  $NPP > NPN$. In that case, we take the rate of nitrogen fixation, $NF^{l,h}$, to be

$$NF^{l,h} = NF_0\left[\exp(NPP^{l,h}/NPN^{l,h}-1)-1\right] \qquad (3)$$

where $NF_0$ has been chosen to be $1\times10^6$ moles $NO_3$ per second.

In the water column, remineralization takes place through oxidation of organic matter with dissolved oxygen as long as oxygen levels are above a certain minimum value $O_{2,min}$, chosen here to be 3 mmol/m$^3$. Below that value, remineralization is assumed to occur by way of denitrification as long as nitrate levels exceed a certain minimum level $NO_{3,min}$, chosen here to be 0.03 mmol/m$^3$. The oxidation equation for denitrification is taken to be

$$(C_{106}H_{124}O_{38})(NH_3)_{16}(H_3PO_4) + 94.4HNO_3 \rightarrow 106CO_2 + 16NH_3 + 47.2N_2 + H_3PO_4 + 109.2H_2O \qquad (R1)$$



Here, as for oxidation with dissolved oxygen in the original DCESS model (Shaffer et al., 2008), we consider that organic matter formed in the ocean surface layer consists of proteins and lipids in addition to carbohydrates and adopt the mean composition proposed by Anderson (1995). Such a composition requires more oxygen for complete oxidation than carbohydrate alone would require (Anderson, 1995). This explains the enhanced nitrate sink in denitrification compared to standard Redfield stoichiometry (factors 94.4 and 47.2 in Reaction R1 compared with 84.8 and 42.4).

When oxygen concentrations are below $O_{2,min}$ and nitrate concentrations are below $NO_{3,min}$, remineralization is assumed occur by way of sulphate reduction for which the oxidation equation is taken to be

$$(C_{106}H_{124}O_{38})(NH_3)_{16}(H_3PO_4) + 44H_2O + 59SO_4^{-2} \rightarrow 106HCO_3^- + 59HS^- + 16NH_3 + H_3PO_4 + 71H^+ \quad \text{(R2)}$$

For the same reason as above, the sulphate sink is somewhat enhanced compared to standard Redfield stoichiometry (factor 59 in Reaction R2 compared with 53). Note that the carbon oxidation product in sulphate reduction is bicarbonate that, as opposed to the carbon dioxide produced in denitrification, contributes to alkalinity (ALK) as well as dissolved inorganic carbon (DIC). Both reactions show a minor contribution to alkalinity by way of ammonia production. In addition, for sulfate reduction with our organic matter model, the ALK decrease from the production of hydrogen ion exceeds the ALK increase from the production of bisulfide ( $HS^-$; from the definition of alkalinity, $ALK \propto HCO_3^- + NH_3 + HS^- - H^+$ (Dickson, 1981)). In summary, for our organic matter stoichiometry, we find ALK/DIC = 0.151 and 1.038 for denitrification and sulfate reduction, respectively.

As shown in Reactions R1 and R2, $NH_3$ and $HS^-$ are produced by denitrification and/or sulphate reduction. As we show below, additional bisulfide is produced by microbes for anoxic conditions by way of anoxic methane oxidation (AMO). In the model, these species are oxidized to nitrate and sulphate, respectively, when transported by advection and diffusion to oxygenated ocean layers where $O_2 \geq O_{2,min}$. These oxidation reactions are

$$NH_3 + 2O_2 \rightarrow NO_3^- + H^+ + H_2O \quad \text{(R3)}$$

and

$$HS^- + 2O_2 \rightarrow SO_4^{2-} + H^+ \quad \text{(R4)}$$

Note that each of the reactions leads to an ALK decrease from production of hydrogen ions.





Oxygenation rates, $\Gamma_{ox,n}^{l,h}$ (in moles per second), are expressed by

$$\Gamma_{ox,\mathrm{NH}_4,n}^{l,h}, \Gamma_{ox,\mathrm{H}_2\mathrm{S},n}^{l,h} = -V_n^{l,h}([\mathrm{NH}_3]_n^{l,h}, [\mathrm{HS}^-]_n^{l,h}) / \tau_{ox,NS} \tag{4}$$

where $n$ is the layer number, $V_n^{l,h}$ are the layer volumes, $[\mathrm{NH}_3]_n^{l,h}$, $[\mathrm{HS}^-]_n^{l,h}$ are the layer species concentrations and $\tau_{ox,NS}$ is an ocean lifetime taken to be 200 days.

The simple ocean nitrogen and sulphur chemistry sketched above is intended to capture the most important processes that
would occur in the ocean following a massive injection of methane without going into further detail of the N and S cycles. Processes that are omitted here include anerobic ammonium oxidation (anammox), oceanic production of nitrous oxide and iron sulphide formation (Kuypers et al. 2003; Freing et al. 2012; Morse et al. 1987). Anammox has been shown to be an important sink of nitrogen nutrient in the ocean but denitrification may still be the most important player here (Ward et al., 2009; Zehr and Kudela, 2011). Treatment of anammox and nitrous oxide would involve in depth consideration of nitrogen
cycling including nitrite production and consumption as well as approaches that transcend our simple, horizontally-averaged model geometry. This is beyond the scope of the present paper but is being addressed by our group in other DCESS model developments in progress. Likewise, sulphide formation would require an explicit treatment of iron cycling also beyond the scope of the model at present.

**3.2.2 Methane oxidation and air-sea gas exchange**

We now include methane as a tracer in the ocean model acted upon by advection, diffusion, air-sea gas exchange and microbial oxidation in the water column. Here we ignore any weak methane production that may take place in the surface ocean in anoxic microenvironments (Reeburgh, 2007) again with the goal of concentrating on massive methane injections.
Oxidation of methane in oxygenated ocean layers ($\mathrm{O}_{2,n}^{l,h} \geq \mathrm{O}_{2,\min}$) produces carbon dioxide and consumes oxygen according to

$$\mathrm{CH}_4 + 2\mathrm{O}_2 \rightarrow \mathrm{CO}_2 + 2\mathrm{H}_2\mathrm{O} \tag{R5}$$

Layer oxidation rates of methane under these conditions, $\Gamma_{ox,\mathrm{CH}_4,n}^{l,h}$, are taken to be:





$$\Gamma_{\mathrm{ox,CH_4},n}^{l,h} = -V_n^{l,h}([\mathrm{CH_4}]_n^{l,h} / \tau_{ox,\mathrm{CH_4}} \tag{5}$$

where $[\mathrm{CH_4}]_n^{l,h}$ are the layer methane concentrations and $\tau_{ox,\mathrm{CH_4}}$ is an ocean methane lifetime for oxic conditions. From a joint analysis of ocean methane and chlorofluorocarbon data, Rehder et al., (1999) found support for modelling methane oxidation as a first order process (Eq. 5) and determined an oceanic methane lifetime of ~ 50 years. Thus we take $\tau_{ox,\mathrm{CH_4}}$ to be 50 years.

For suboxic/weakly anoxic conditions characterized by $\mathrm{O_2} < \mathrm{O_{2,min}}$ and $\mathrm{NO_3} > \mathrm{NO_{3,min}}$, we adopt the following reaction for nitrate-dependent, microbial AMO (Raghoebarsing et al. 2006; Cui et al. 2015)

$$5\mathrm{CH_4} + 8\mathrm{HNO_3} \rightarrow 5\mathrm{CO_2} + 4\mathrm{N_2} + 14\mathrm{H_2O} \tag{R6}$$

An alternative and slightly more energetic reaction involves nitrite, a species that we however do not consider in our simplified nitrogen cycling approach (Haroon et al. 2013).

For anoxic conditions for which $\mathrm{O_2} < \mathrm{O_{2,min}}$ and $\mathrm{NO_3} < \mathrm{NO_{3,min}}$, sulphate-dependent, microbial AMO (Treude et al., 2005; Cui et al., 2015) may be expressed as

$$\mathrm{CH_4} + \mathrm{SO_4^{-2}} \rightarrow \mathrm{HCO_3^-} + \mathrm{HS^-} + \mathrm{H_2O} \tag{R7}$$

Note that whereas nitrate-dependent, microbial AMO (Reaction R6) does produce ALK, the production of bicarbonate and bisulphide in sulphate-dependent, AMO (Reaction R7) is a strong ALK source leading to an ALK/DIC ratio of 2, considerably higher than for sulphate reduction of organic matter (Reaction R2). From carbonate chemistry, an increase in ocean alkalinity tends to depress dissolved carbon dioxide concentrations in the ocean and, by way of air-sea gas exchange, also in the atmosphere.

Layer oxidation rates of methane for sulphate- dependent AMO, $\Gamma_{\mathrm{anox,CH_4},n}^{l,h}$, are taken to be:

$$\Gamma_{\mathrm{anox,CH_4},n}^{l,h} = -V_n^{l,h}([\mathrm{CH_4}]_n^{l,h} / \tau_{anox,\mathrm{CH_4}} \tag{6}$$





where $\tau_{anox,CH_4}$ is an ocean methane lifetime for anoxic conditions ($O_2 < O_{2,min}$ and $NO_3 < NO_{3,min}$) that may be considerably longer than $\tau_{ox,CH_4}$ since Reaction R7 is much less energetic than Reaction R5 (Lam and Kuypers, 2011). To take this into consideration and to be consistent with much lower anoxic remineralization rates compared to oxic rates in the DCESS model sediment module (Shaffer et al., 2008) , we adopt $\tau_{anox,CH_4} = 0.1\tau_{ox,CH_4}$ or 500 years as our standard case. On

the other hand, nitrate-dependent AMO (Reaction R6) is energetically similar to oxic methane oxidation (Reaction R5) so we take $\tau_{ox,CH_4}$ for the methane lifetime associated with nitrate-dependent AMO.

Air-sea gas exchange for methane, $\Psi_{CH_4}^{l,h}$ , is calculated as

$$\Psi_{CH_4}^{l,h} = k_w^{l,h}(\eta_{CH_4}^{l,h} pCH_4 - [CH_4]^{l,h}) \tag{7}$$

where the gas transfer velocities $k_w^{l,h}$ are $0.39(u^{l,h})^2 Sc_{CH_4}^{l,h}/660)^{0.5}$ whereby $u^{l,h}$ are mean wind speeds at 10 m above the ocean surface, taken to be 8 m s$^{-1}$ in both zones, and $Sc_{CH_4}^{l,h}$ are Schmidt numbers for methane that depend on ocean surface layer temperature (see below). The methane solubility, $\eta_{CH_4}^{l,h}$ , was converted from the Bunsen solubility coefficients that

depend on surface layer temperature and salinity (Wiesenburg and Guinasso, 1979) to model units using the ideal gas mole volume. Furthermore, $[CH_4]^{l,h}$ is the dissolved methane concentration in the ocean surface layers.

In recent work it was shown that the traditional calculations of $Sc_{CO_2}$ based on a third order polynomial of surface layer temperature (Wanninkhof, 1992) seriously underestimated $Sc_{CO_2}$ for temperatures ($T$) above 30°C (Gröger and

Mikolajewicz , 2011). These authors proposed the form $Sc = a\exp(-bT) + c$ and fit that equation using experimental data of Jähne et al., (1987) to find best fit values for $CO_2$ of 1956, 0.0663 and 142.6 for *a, b* and *c,* respectively. Since we are dealing here with simulations of extreme global warming, we adopt this new $Sc_{CO_2}$ expression and fit here, thereby superseding the traditional $Sc_{CO_2}$ calculation previously used in our model (Shaffer et al., 2008). Gröger and Mikolajewicz (2011) did not consider methane but we directly fit their proposed exponential function to the experimental data of Jähne et

al., (1987) for $Sc_{CH_4}$ over 5°C intervals in the temperature range $0 - 40$°C. We found best fit values ($R^2 = 0.997$) for *a, b* and *c*, respectively, such that $Sc_{CH_4} = 1771\exp(-0.0650T) + 128.8$ . We adopt this relationship here.



## 4 Sample simulations with methane cycling

### 4.1 Methane input specification

The rate of methane input, $F_{CH_4}$, we apply to the atmosphere/ocean system is taken to be

$$F_{CH_4}(t) = 404 F_{CH_4}^{tot} / (\tau_{in})^5 \exp(-6.27t/\tau_{in}) \tag{8}$$

where $F_{CH_4}^{tot}$ is the total input in GtC, $\tau_{in}$ is a time scale in years over which the first 75% of the input is added and $t$ is the

time in years. This mathematical form has been used before within our group to characterize $CO_2$ input (Shaffer 2009; Shaffer et al., 2016). As discussed in the introduction, the great bulk of both possible major $F_{CH_4}$ sources to fuel deep time warming events, MH and TM, are likely to enter via the ocean. Estimates of the time scale for the carbon input driving the deep time global warming events have varied from years to tens of thousands of years (eg. Zeebe et al., 2009; Cui et al., 2011; Wright and Schaller, 2013). In the lower end of this time scale range, Earth System response to $F_{CH_4}^{tot}$ will vary

significantly depending on whether this methane is dissolved in the ocean or part or even all of it escapes as gas to the atmosphere, in bubble plumes for example (McGinnis et al., 2006). Such response variation may be explained in part by much shorter time scales for methane oxidation to carbon dioxide (in both the atmosphere and ocean) than ocean diffusive and overturning time scales. Escape to the atmosphere will cause immediate global warming since methane is a powerful greenhouse gas, much more so than carbon dioxide (Myhre et al., 1998). Oxidation of dissolved methane in the ocean will

consume dissolved oxygen there (Reaction R5). For oxidation of large amounts of methane, suboxic or anoxic conditions may develop with complex nutrient and carbon cycling consequences upon which we elaborate below. We address these questions here by taking the direct, undissolved methane escape to the atmosphere to be $\alpha F_{CH_4}$ and the methane dissolution in the ocean to be $(1-\alpha)F_{CH_4}$, whereby $\alpha$ is the fraction of total input that directly enters the atmosphere. Furthermore, we must specify how methane dissolution is distributed between the two ocean sectors and among the 55 layers of each sector.

Here we simply assume that 84% of the dissolution occurs in the low-mid latitude sector and 16% in the high latitude sector, a division that reflects model sector areas. Furthermore, we assume that the amount assigned to each sector is divided equally among the upper 30 model ocean layers, spanning the depth ranges over which MH and/or TM inputs may be expected (e.g. Buffett and Archer, 2004; Svensen et al., 2004)..





## 4.2 Pre-event steady state and a long sample simulation

In order to investigate the impacts of these model improvements on simulation results, we chose as an initial case study the
late Paleocene (pre-PETM) configuration and associated model steady states constructed in Shaffer et al. (2016) but now
including some changes like the adaption of the Byrne and Goldblatt (2014a) expressions for carbon dioxide radiative
forcing and for carbon dioxide-nitrous oxide overlap. In this configuration we assume a solar constant decrease of 0.5 %
from present day, sea level rise of 100 m, mean ocean salinity of 33.8, slightly reduced background albedo (to represent less
land in the subtropics and more vegetation cover), extra high latitude, longwave cloud forcing of 30 W/m$^2$, PI atmospheric
oxygen level, adjustments in ocean calcium and magnesium concentrations and adjustments in initial lithosphere outgassing
and weathering inputs. Furthermore, parameters in the model's simplified treatment of outgoing long wave radiation were
adjusted to yield a global mean temperature of 25 °C and a climate sensitivity in the range 4 -5 °C for atmospheric $p$CO$_2$
levels in the range 800-900 ppm (Shaffer et al., 2016). Open system steady states were then sought (by iteration and/or long
time integration) such that external inputs (lithosphere outgassing and weathering) balanced external output (burial down out
of the ocean sediment). Table 1 lists some properties of the model steady state we chose among various possibilities given
the above constraints. This particular model configuration with a global mean temperature of 25.0 °C and an atmospheric
$p$CO$_2$ of 860 ppm has a climate sensitivity of 4.8 °C. While this steady state largely reflects late Paleocene conditions to
ease comparison with our prior work, we emphasize that a similar approach can be taken to design appropriate initial
conditions for any particular deep time warming event, like the End-Permian event (Shen et. al., 2011), based on paleo-
reconstructions for conditions prior to that particular event.

Figure 4 shows time evolutions of some relevant model variables from a 200,000 year simulation that started in equilibrium
from the above initial conditions and was forced by methane input of 4000 GtC over a time scale ($\tau_{in}$) of 3 kyr, whereby
half of the input reaches the atmosphere and half is dissolved in the ocean ($\alpha = 0.5$). Furthermore, to be specific it is
assumed that the δ$^{13}$C of the methane input is -40 ‰. Such forcing leads to maximum event temperature and $p$CO$_2$ rises of
5.7 °C and 1040 ppm, a peak atmospheric $p$CH$_4$ of 10.3 ppm and a low-mid latitude, Calcite Compensation Depth (CCD)
shoaling of 463 m (Fig. 4a,c,h).

The low-mid latitude ocean warms by up to 5.5 °C at the surface, increasing to 6.0 °C at depth, reflecting modest polar
amplification of the warming by about 20% (Fig. 4b). Mean low-mid latitude dissolved O$_2$ concentrations at 1000 m depth
approach zero after less than 2 kyr and remain so for an additional 3 kyr, driven to a large extent by oxidation of methane
dissolved in the ocean (Fig. 4d). Considerable denitrification occurs during this 3 kyr long, anoxic period, reflected in an
associated decrease in new production due to nitrogen limitation (Fig. 4e). After this decrease, ocean new production then
steadily increases to about 12% higher than pre-event levels by 24 kyr (due to more ocean phosphate from enhanced

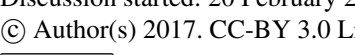


weathering) and then slowly decreases again as weathering decreases with temperature. There is only a slight initial decrease of about 100 GtC in land biosphere carbon inventory as more/less new production from $CO_2$ fertilization was roughly balanced by more/less soil remineralisation from warming/cooling (Fig. 4f). The initial decrease in both ocean new production and land biomass enhance atmospheric $pCO_2$ and warming while the subsequent ocean new production increase
followed by a slow decrease modulate the slow decrease in $pCO_2$ and slow cooling following peak warming.

Weathering increases over the event are accompanied by an initial drop in carbonate burial as dissolution decreases the amount of sediment $CaCO_3$ as well as the burial velocity (Fig. 4g,h). There is little initial change in organic carbon burial as the opposing effects tend to balance: decreasing burial velocity and increased organic matter preservation from decreasing
$O_2$ concentrations. During this period there is a net source of carbon to the ocean-atmosphere system (in addition to the prescribed methane input) as volcanic/weathering inputs exceed burial outputs. After the methane input event, both $CaCO_3$ and organic carbon burial increase in response to higher new and $CaCO_3$ production. Together with decreasing weathering in response to cooling, this results in weak but long-lived net sink of carbon from the ocean-atmosphere system (Fig. 4g). In combination with decreasing $pCO_2$ and associated increasing ocean $[CO_3^{2-}]$, the biogenic $CaCO_3$ production increase drives
an overshoot of the low-mid latitude CCD to depths greater than the pre-event depth, in agreement with theory and sediment core data (Dickens et al., 1997; Leon-Rodriguez and Dickens, 2010; Penman et al. 2016).  This overshoot, that is even more pronounced at high latitudes, peaks by 35 - 45 kyr into the simulation (Fig. 4h).

The low-mid latitude, pelagic $\delta^{18}O$ excursion in biogenic $CaCO_3$ is significantly muted compared to the benthic one due to
enhanced $\delta^{18}O$ in the surface layer ($\delta^{18}O_w$) from a more vigorous hydrological cycle during the warming event (Fig. 4i) The carbon isotope excursion (CIE) is slightly muted in the atmosphere and enhanced in the low-mid latitude, ocean surface layer due to warming and temperature-dependent fractionation in air-sea gas exchange (Fig. 4j). The low-mid latitude, marine organic matter CIE is greater still, in agreement with paleo-reconstructions (McInerney and Wing, 2011), as a consequence of model dependence of carbon isotope discrimination during ocean photosynthesis on concentrations of dissolved carbon
dioxide and phosphate in the ocean surface layer (Shaffer et al., 2016). Use of a $pCO_2$-dependent carbon isotope fractionation leads to a still greater CIE in terrestrial organic matter, also in agreement with paleo-reconstructions (Jahren and Schubert, 2013).  Note that the model CIE is more protracted than the initial warming or the $\delta^{18}O$ excursion related to the warming (Fig. 4a,i,j). Relaxation back to about half of maximum CIE values takes place over about 80 kyr compared to about 40 kyr for comparable temperature or $\delta^{18}O$ relaxation. This can be explained by enhanced burial of $\delta^{13}C$-enriched
$CaCO_3$ after the methane input event. After 200 kyr, most of model Earth System properties have relaxed back toward pre-event values. This long time scale is dictated by external input/output balances of the global carbon cycle (Fig. 4g), in particular by the slow, negative silicate weathering feedback on climate (Shaffer et al., 2008).



### 4.3 Sensitivity to input size, time scale and distribution

Figure 5 shows selected model results for the first 20,000 years of simulations like that of Fig. 4 but with methane input amounts of 2000, 4000 and 6000 GtC. Both $pCO_2$ and especially $pCH_4$ increase more than linearly in response to the linear increase in methane input (Fig. 5a,c,d). For $pCO_2$ this is due to positive feedbacks from the land biosphere and ocean. Initial warming from the methane injection leads to greater soil remineralization, land carbon inventory decrease and input of $CO_2$ to the atmosphere (Fig. 5f). Increasing methane oxidation in the ocean leads to denitrification that draws down ocean new production leading to increased ocean $CO_2$ outgassing that was already enhanced in response to decreased ocean $CO_2$ solubility from warming (Fig. 5e). The non-linear increase in $pCH_4$ is due mainly to the increase in $CH_4$ atmospheric lifetime with $pCH_4$ (Fig. 2). On the other hand, warming increases less than linearly (Fig. 5b) as expected from the radiative forcing dependencies shown in Fig. 3 and from our neglect here for simplicity of climate sensitivity increase with warming (Shaffer et al., 2016). Net carbon input from volcanism/weathering minus burial follows process sequences as described for Fig. 4 above (Fig. 5g). The CCD shoals as more methane is oxidized to $CO_2$ in the atmosphere and ocean leading to more acidic conditions that drive calcite dissolution in the sediment (Fig. 5h). However, CCD shoaling increase is damped for large methane dissolution in the ocean since this leads to anoxic conditions, methane oxidation with sulphate (Reaction R7) and associated alkalinity inputs that oppose dissolution. This also has had the effect of damping the atmospheric $pCO_2$ increase. In response to warming, oxygen isotopes of the low-mid latitude ocean surface water are enriched by enhanced evaporation associated with a stronger atmospheric hydrological cycle (Fig. 5i). Low-mid latitude, ocean surface layer salinity (not shown) increases in response to warming for the same reason and the salinity excursion can be approximated very well by multiplying the results in Fig. 5i by 2.8. Atmospheric CIE values are directly proportional to the size of the methane input and exhibit a long time decay scale as noted in Fig. 4 above (Fig. 5j):

Figure 6 shows selected model results for the first 20,000 years of simulations like that of Fig. 4 but with methane injection times $\tau_{in}$ of 300, 1000, 3000 and 10,000 years. For the shortest injection times, input rates in terms of carbon equal or exceed those of present day anthropogenic carbon emissions exceeding 10 GtC/yr (Fig. 6a; IPCC, 2013). Global warming spikes to much higher values for short injection times in response to large instantaneous methane radiative forcing and under the influence of slow ocean exchange times (Fig. 6b,d). Warming peaks at 8.0, 6.7, 5.7 and 4.8°C for the shortest to the longest injection times. Atmospheric $pCO_2$ is also enhanced for shorter injection times in part due to the positive feedbacks from the land biosphere and ocean noted above (Fig. 6c,e,f). Shorter injection times lead to more widespread sub-oxic/anoxic conditions as evidenced by initial decreases in ocean new production in response to denitrification (Fig. 6e). These conditions are a direct response to increased methane oxidation in the ocean due to enhanced methane input rates (Fig. 6a). One consequence of this situation is the muted CCD shoaling for shorter injection times following from enhanced methane oxidation with sulphate (Reaction R7) and associated alkalinity inputs that oppose $CaCO_3$ dissolution (Fig. 6h).





Both net carbon input and especially oxygen isotopes from low-mid latitude, ocean surface layer water follow global warming due in large part to climate-dependent weathering and temperature-dependent, atmospheric water vapor transport (Fig. 6a,g,i; Shaffer et al., 2008). The atmospheric CIE is much enhanced for short injection times. This can be understood as follows: the carbon isotope signal from the methane injected to the atmosphere, and transferred there to $CO_2$ via oxidation

of the methane, builds up in the atmosphere and in the ocean surface layer (not shown) due to long exchange times of about 1 kyr with the deep ocean (Fig. 6j). The atmospheric CIE tops out at -5.24, -3.80, -3.21 and -3.03 for the shortest to the longest injection times.

Figure 7 shows selected model results for the first 20,000 years of simulations like that of Fig. 4 but with 0, 50 and 100% of

the methane input reaching the atmosphere ($\alpha = 0$, 0.5 and 1). Global warming, atmospheric $pCH_4$ and, to a lesser extent, atmospheric $pCO_2$ are enhanced for more input directly to the atmosphere (Fig. 7b,c,d). Warming is enhanced by 1.7°C for all methane input to the atmosphere compared to all input dissolved in the ocean. Even with all methane input dissolved in the ocean, atmospheric $pCH_4$ increases initially by more than 2 ppm as methane outgases from the ocean via air-sea gas exchange (Fig. 7d). The positive land biosphere feedback on atmospheric $pCO_2$ is enhanced for all methane to the

atmosphere due to enhanced warming (Fig. 7f). On the other hand, ocean feedbacks on atmospheric $pCO_2$ are more nuanced. The positive, solubility feedback is enhanced for all methane to the atmosphere due to enhanced warming whereas the positive feedback from reduced ocean new production is enhanced for all methane to the ocean (Fig. 7e). The latter effect is due to methane oxidation in the ocean leading to denitrification and nitrogen limitation of new production. Enhanced $CaCO_3$ dissolution in the ocean sediment is a further consequence of methane oxidation in the ocean when all methane is dissolved

there. This leads to decreased calcite burial and enhanced net carbon input to the ocean-atmosphere system despite less weathering input from less global warming for this case (Fig. 7g). This also explains the enhanced CCD shoaling for all methane dissolved in the ocean (although modulated somewhat from enhanced methane oxidation with sulphate and associated alkalinity inputs as discussed above; Fig. 7h). Again, oxygen isotopes from low-mid latitude, ocean surface layer water follow global warming due to temperature-dependent, atmospheric water vapor transport (Fig, 7i). The atmospheric

CIE peaks slightly earlier for all methane to the atmosphere but, due to the long time scale of the methane input (3000 kyr) relative to ocean overturning times, the CIE amplitude is essentially the same for all cases (Fig. 5j).

### 4.4 Modelled ocean distributions

Figure 8 shows selected, low-mid latitude model ocean results for the first 20,000 years of a simulation forced by a methane input of 6000 GtC with $\delta^{13}C$ = -40 ‰ over a time scale ($\tau_{in}$) of 3 kyr, whereby all the input is dissolved in the ocean ($\alpha = 0$). This forcing configuration was chosen to highlight the workings of the model for suboxic/anoxic conditions and leads to maximum event temperature and $pCO_2$ rises of 5.7 °C and 1095 ppm, a peak atmospheric $pCH_4$ of 6.2 ppm and a low-mid latitude CCD shoaling of 389 m (not shown).



Low-mid latitude ocean warming is greatest at depth, a reflection of polar amplification (Fig. 8a). Maximum warming there (6.2 °C) occurs about 1500 years after maximum surface layer (or atmosphere) warming. Subsurface methane concentrations build up to more than 30 mmol/m$^{-3}$ (Fig. 8b). Oxidation of this methane with dissolved oxygen (Reaction R5) forces suboxic

conditions ($O_2 < 10$ mmol/m$^{-3}$) for thousands of years at intermediate and mid-depths of the low-mid latitude ocean (Fig. 8c). Furthermore, a combination of reduced solubility due to warming and enhanced remineralization due to increased ocean new production (Figs. 5-7) maintain suboxic conditions at intermediate depths there well beyond the 20,000 simulation years shown here. As dissolved oxygen is forced below $O_{2,min}$ (3 mmol/m$^3$), methane oxidation proceeds by way of nitrate-dependent, anoxic methane oxidation (Reaction R6), essentially eliminating nitrate for several thousand years at intermediate

and mid-depths of the low-mid latitude ocean (Fig. 8d). Over a period of several thousand years after most methane is oxidized, nitrate concentrations recover to and even above pre-event values. This can be explained by a combination of nitrogen fixation (Eq. 3) and enhanced new production due to increased ocean phosphate concentrations from warming-enhanced weathering. As nitrate is forced below $NO_{3,min}$ (0.03 mmol/m$^3$), methane oxidation proceeds by way of sulphate-dependent, anoxic methane oxidation (Reaction R7). This leads to the production of hydrogen sulphide that reaches

concentrations over 10 mmol/m$^{-3}$ at intermediate depths almost 3 kyr into the simulation (Fig. 8e).

Fig. 8f shows the model low-mid latitude, time-space distribution of Omega ($\Omega$) defined as the ratio of carbonate ion concentration to carbonate ion saturation concentration for calcite. Calcite dissolution in the sediment increases as $\Omega$ decreases from 1 and model biogenic calcite production in the surface layer is proportional to $(\Omega-1)/\{1+(\Omega-1)\}$ for $\Omega \geq 1$

and zero for $\Omega < 1$ (Shaffer et al., 2016). In response to methane oxidation to $CO_2$, carbonate ion and thereby $\Omega$ initially decrease in intermediate and mid-depths where this oxidation occurs. However, as dissolved oxygen and nitrate are consumed and sulphate-dependent AMO takes over about 2 kyr into the simulation, $\Omega$ increases in these layers in response to alkalinity produced in this reaction. After the period of sulphate-dependent AMO, $\Omega$ decreases again followed by a slow increase driven by decreasing $CO_2$ levels from carbonate compensation. Low-mid latitude, oxygen isotope excursions in

biogenic carbonate produced in-situ (Fig. 8g) track temperature changes in the deep ocean but are reduced relative to these changes near the surface due to enhanced ambient water isotopes (Fig. 7i). After several thousand years, low-mid latitude, carbon isotope excursions in biogenic carbonate produced in-situ (Fig. 8h) are slightly depressed in the deep ocean. This is due to dilution from additional, less-depleted carbon inputs associated with sediment $CaCO_3$ dissolution.

Figure 9 shows selected, low-mid (LL) and high latitude (HL) model ocean profiles at 0, 3 and 5 kyr into the simulation of Fig. 8 and serves to illustrate differences between the LL and HL ocean zones. Although methane input is scaled to be comparable per unit area for both zones, much higher methane concentrations build up in the LL zone than in the HL zone (Fig. 9a). Much higher HL vertical exchange spreads methane vertically and promotes some outgassing to the atmosphere there. Phosphate concentrations increase with time in response to warming-enhanced weathering (Fig. 9b), explaining the



slow increase in ocean new production shown above (Figs. 5-7). Despite sub-oxic and eventually anoxic conditions in the LL zone, the HL zone remains oxygenated due to stronger vertical exchange and the downwelling branch of the overturning circulation there (Fig. 9c). As a consequence, HL nitrate concentrations remain elevated and there is no anoxic remineralization nor anoxic methane oxidation there (Fig. 9d). This is also explains the essential lack of HL ammonium and

hydrogen sulphate, $< 0.02$ and $< 0.1$ mmol/m$^3$, respectively (Figs 9e,f). Ammonium or hydrogen sulphate reaching the HL zone via horizontal exchange with the LL zone is quickly oxidized. By way of carbonate chemistry, addition of $CO_2$ to the water column via methane oxidation depresses carbonate ion concentrations (Fig. 9g). In terms of $\Omega$ (see above) this reduction is relatively stronger in the HL surface layer, tending to depress biogenic carbonate production there. However, this reduction is largely offset by warming enhancement. Addition of $CO_2$ to the water column via methane oxidation

increases ocean acidity (Fig. 9h). For the simulation considered here, surface layer pH is reduced by 0.26 and 0.28 for the LL and HL zones, respectively, from pre-event levels already 0.47 and 0.48 lower than model PI levels (Shaffer et al., 2008).

### 4.5 Modelled ocean sediment properties and synthetic sediment cores

DCESS model simulations include prognostic values for burial velocities down out of each of the model's 110 bioturbated sediment layers, each 10 cm thick, as well as sediment properties exported down out of these layers. In the original model, synthetic sediment cores (SSC) produced in this way only contained information on calcite and organic matter content of the buried sediment (Shaffer et al., 2008). We have extended this treatment to include carbon and oxygen isotopes and report the first results of this here for bulk carbonate, in the form of calcite in the model. For each sediment layer we consider

conservation of calcite as well as conservation of carbon and oxygen isotopes in this calcite. This involves tracking the time-dependent inputs (from the rain of biogenic calcite produced in the ocean surface layer) and outputs (dissolution within the sediment layer and export down out the layer by burial). Isotope effects for changes in surface layer $CO_3^{2-}$ are applied in the production of biogenic calcite (Spero et al., 1997). The new model extension also accounts for effects of possible "mining" of buried sediment (i.e. upward directed "burial" velocities) in response to very strong dissolution events (Shaffer et al.,

2008). For this, properties buried in each SSC are recorded to provide correct values for properties reentering the active sediment layer from below during any such "mining" event.

Figure 10 shows bulk carbonate results from model low-mid and high latitude SSCs at 3000 m depth together with ocean surface layer properties, all for the simulation of Figure 8. Burial velocities and sediment calcite content are greater in the

HL zone compared to the LL zone due to considerably greater model HL rain rates per unit area (Figs. 10a,b). This can be explained in part by temperature-enhanced, HL rain rates in the model: the pre-event HL ocean surface layer is almost 10°C warmer than for the PI simulation (Shaffer et al., 2008). For this reason and slightly higher new production (Table 1), pre-event, HL biogenic calcite rain is more than three times greater than for in the PI case. On the other hand, pre-event, LL biogenic calcite rain is only about 15% greater than for the PI case. Over the first 20 kyr of the simulation shown here, 38.7





cm of sediment is laid down at 3000 m depth in the HL zone compared to 14.0 cm in the LL zone. Burial velocities and sediment calcite content decrease initially in response to increased dissolution and, to a lesser extent, decreased rain rate, as a consequence of the methane input and its oxidation in the ocean. Later on, both properties increase to exceed their pre-event levels in response to increasing new production from increasing ocean phosphate levels.

In both model zones the carbon isotope excursion in the SSC bulk carbonate is modestly attenuated relative to the ocean surface layer excursions (Figs. 10c,e). This is due to reservoir time scales of up to several thousand years for the active sediment layer and, to a lesser extent, to the $CO_3^{2-}$ isotope effect (a reservoir time scale is the calcite inventory of a sediment layer divided by rate of calcite input to it). The reservoir effect also leads to somewhat distended bulk carbonate excursion

maxima centered 3-4 kyr after surface layer maxima (note that the time axis of the figure is "stretched/squeezed" relative to the sediment length axis, a product of time-varying burial rates). Similar conclusions and interpretations also apply to the oxygen isotope excursions in the SSC bulk carbonate and the ocean surface layers in both model zones (Figs. 10d,f). Note that if, in lack of further knowledge, a linear time scale would be assigned to SSC length, the bulk carbonate excursions would appear sharper and shorter, particularly in the HL zone. When used as a paleothermometer, SSC bulk carbonate in the

LL zone severely underestimates the model temperature excursion in the LL surface layer, primarily due to the positive ambient water $\delta^{18}O$ excursion demonstrated above, but with a minor contribution from the $CO_3^{2-}$ isotope effect (Fig. 10d). On the other hand, the slightly negative ambient water $\delta^{18}O$ excursion in the HL surface layer and the $CO_3^{2-}$ isotope effect there work to improve the estimate of the surface layer warming from the $\delta^{18}O$ excursion of SSC bulk carbonate, albeit with the 3-4 kyr time lag due to the reservoir effect (Fig. 10f).

## 5. Discussion and conclusions

Here we have extended the DCESS Earth System model to include methane cycling. This is a necessary step for dealing with deep time global warming events, some of which correspond to major life extinction events, since most of these warming

events were probably forced by massive methane inputs (Berner, 2002; Hesselbo et al., 2000; McElwain et al., 2005; Retallack & Jahren, 2008; Ruhl et al., 2011; Svensen, H. et al., 2004; Shaffer et al., 2016). To be able to treat impacts of such inputs more correctly and consistently, we have also extended the model to deal with suboxic/anoxic conditions in the ocean and their consequences. For this we have now included denitrification, nitrogen fixation, sulphate reduction and nitrate/sulphate-dependent, anoxic methane oxidation. Furthermore, we have upgraded the treatment of methane for high

concentrations in the model atmosphere with the latest radiative forcing relationships and with improved relationships for atmospheric lifetimes. To our knowledge, no other Earth System model of any degree of complexity has yet been formulated that can deal as comprehensively with extreme methane inputs and their Earth System consequences.



After formulating the model extensions we embark on extensive tests of model behaviour for methane inputs of various sizes, time scales and locations. We demonstrate model behaviour over event time scales exceeding 100 kyr but concentrate on the first 20 kyr of the simulations. The long, > 100 kyr simulation demonstrates how warming-driven weathering increases lead to a slow buildup of ocean phosphate concentrations and ocean new production. This modulates the evolution of atmospheric $p$CO$_2$ and helps to explain model CCD deepening overshoot after the initial CCD shoaling from the methane input. Extensive ocean anoxia develops for larger methane inputs over shorter time scales with more methane dissolving in the ocean. For such anoxia, there is much sulphate-dependent, anoxic methane oxidation that produces hydrogen sulphide but also alkalinity that works to oppose calcite dissolution in the sediment and $p$CO$_2$ rise in the atmosphere. Furthermore, extensive denitrification also occurs that initially depresses ocean new production, leading to $p$CO$_2$ outgassing, until nitrogen fixation steps in to fill the nitrate gap. The global warming excursion is greater when more methane escapes to the atmosphere, leading to higher $p$CH$_4$ and more radiative forcing there. Initial, methane-driven warming forces increased soil remineralization and CO$_2$ input to the atmosphere until subsequent $p$CO$_2$ rise and associated CO$_2$ fertilization turns the tables. Carbon and oxygen isotope excursions of bulk biogenic carbonate in model synthetic sediment cores (SSC) are attenuated, distended and delayed relative to carbon and oxygen isotope excursions in the ocean surface layer where the carbonate is formed. Oxygen isotope excursions in surface layer water compromise the use of oxygen isotope excursions in model SSC bulk carbonate to gauge surface layer warming in the low-latitude ocean zone. However, such an oxygen isotope-based, paleo-thermometer works well in the high latitude ocean zone.

Methane is the most likely main source of carbon-13 depleted, carbon inputs forcing deep time, global warming events: Not enough organic carbon is available in the land biosphere to supply carbon needed to explain observed carbon isotope excursions and the very large volcanic carbon input needed to explain the observed isotope excursions greatly exceeds carbon inputs consistent with observed CCD shoaling (Shaffer et al., 2016). Such methane input may be in the form of thermogenic methane ($\delta^{13}$C~ -40 ‰) and methane hydrate ($\delta^{13}$C~ -60 ‰). There is an ongoing debate as to whether enough methane hydrate could exist for deep time, warm pre-event conditions (Buffett and Archer, 2004; Gu et al., 2011). To be specific in the present modelling development paper we chose $\delta^{13}$C of our prescribed methane inputs to be -40 ‰. However, more extensive and careful work will be required to resolve this issue. As one step in this direction, some of us are working to develop a methane hydrate module for the DCESS model. In combination with the comprehensive methane cycling implemented in the present paper and in the presence of a large MH reservoir in present ocean sediments, such a module could also be used for improved projections of future, long term warming with the DCESS model, building upon earlier work (Shaffer et al., 2009). It is also possible that different deep time, warming events had different methane sources or the two methane types in some combination (e.g. Freiling et al., 2016). With the present implementation of methane cycling in the DCESS model we are now in the position to systematically assess the amounts, types, time scales and locations of methane injections driving specific deep time, global warming events.



**Model availability**

The DCESS model code can be downloaded at www.dcess.dk.

**Author contributions**

G. Shaffer designed the work, wrote the original draft and led the development of model extensions and code, to which E. Fernandez Villanueva, R. Rondanelli, S. Olsen and J.O.P. Pedersen contributed. E. Fernández Villanueva carried out the model simulations. All authors discussed model concepts and results and contributed to the final manuscript.

**Competing interests**

The authors declare that they have no conflict of interest.

**Acknowledgments**

Some of this work was carried out at the Center for Advanced Studies in Arid Zones (CEAZA), La Serena, Chile. This research was supported by FONDECYT (Chile) grant 1150913 and Chilean ICM grant NC120066.

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

|  | Pre-event | PI |
|---|---|---|
| Atmosphere mean temperature (°C) | 25.0 | 15.0 |
| Atmosphere $p\mathrm{CO_2}$ (ppm) | 860 | 278 |
| Atmosphere $p\mathrm{CH_4}$ (ppm) | 1.51 | 0.72 |
| Atmosphere carbon inventory (GtC) | 1826 | 590 |
| Land carbon inventory (GtC) | 2597 | 2220 |
| Ocean carbon inventory (GtC) | 33,473 | 37,910 |
| Total carbon inventory (GtC) | 37,895 | 40,720 |
| Low-mid latitude CCD depth (m) | 4239 | 4673 |
| Ocean new production (GtC/yr) | 6.47 | 5.40 |
| Ocean biogenic $\mathrm{CaCO_3}$ production (GtC/yr) | 1.47 | 0.97 |
| Ocean mean $\mathrm{O_2}$ (mmol/m$^3$) | 111.2 | 183.5 |

**Table 1: Pre-event steady state solution properties compared to pre-industrial (PI) properties**





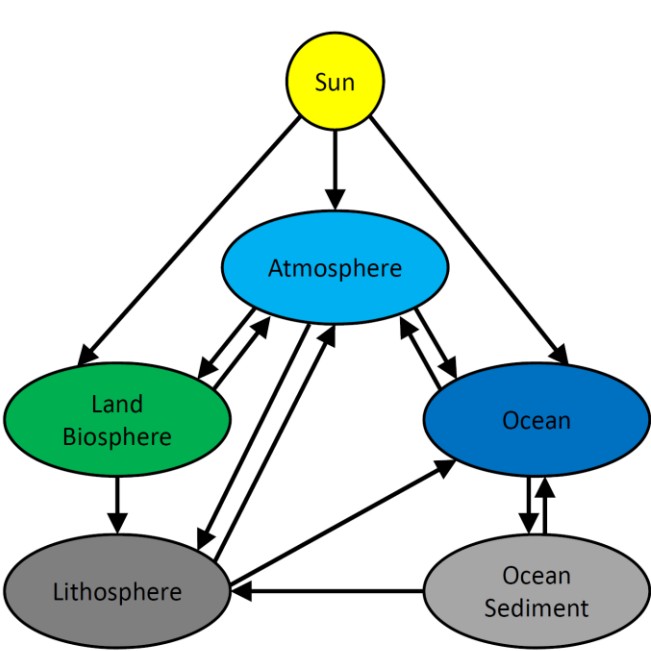

**Figure 1: Modules and interconnections for the Danish Center for Earth System Science (DCESS) Earth System**
15 **model**





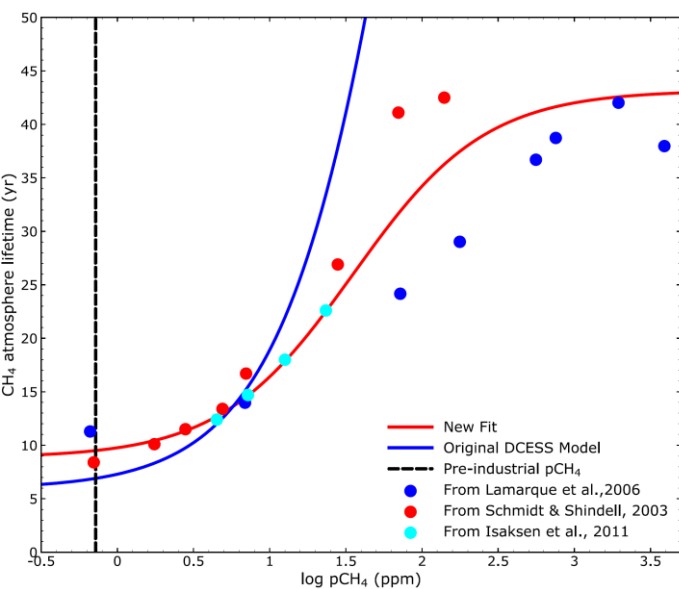

**Figure 2: Atmospheric lifetime of methane as a function of atmospheric methane concentration.** See text for a
10   discussion of the model results and the functional fit to them.



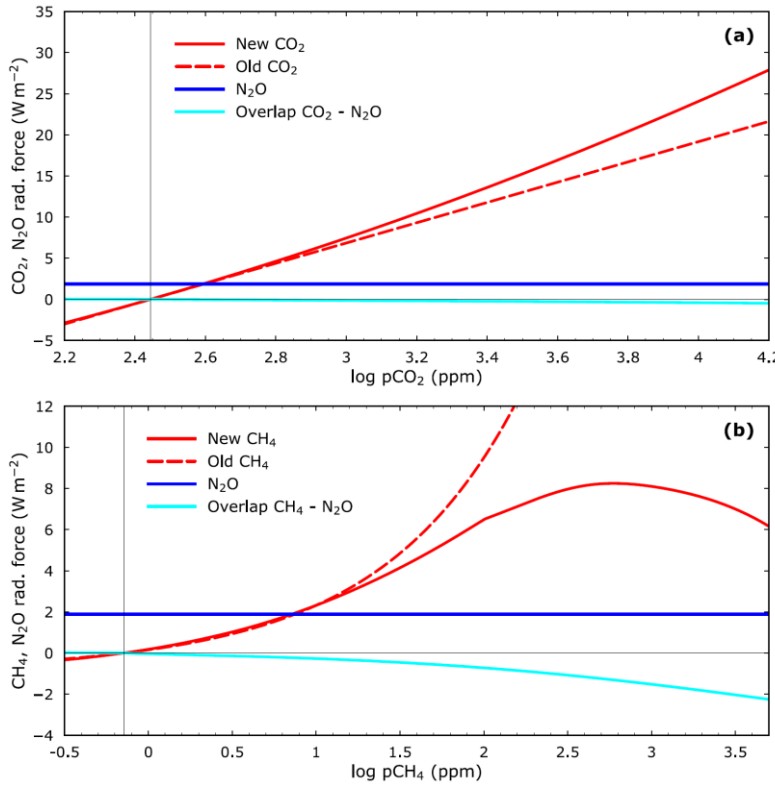

**Figure 3: Radiative forcing of carbon dioxide, methane and nitrous oxide as functions of their atmospheric concentrations. a)** $CO_2$ radiative forcing used here (new $CO_2$; Byrne and Goldblatt, 2014a) compared to $CO_2$ forcing in the original DCESS model (old $CO_2$; Shaffer et al., 2008). $CO_2$-$N_2O$ overlap (Bryne and Goldblatt, 2014a) and $N_2O$ radiative forcing calculated for constant, pre-industrial $N_2O$ partial pressure of 270 ppb., **b)** $CH_4$ radiative forcing used here (new $CH_4$;

10    Byrne and Goldblatt, 2014a,b) compared to $CH_4$ forcing in the original DCESS model (old $CH_4$). $CH_4$-$N_2O$ overlap (Byrne and Goldblatt, 2014a) and $N_2O$ radiative forcing calculated for constant $N_2O$ partial pressure as in **a**.





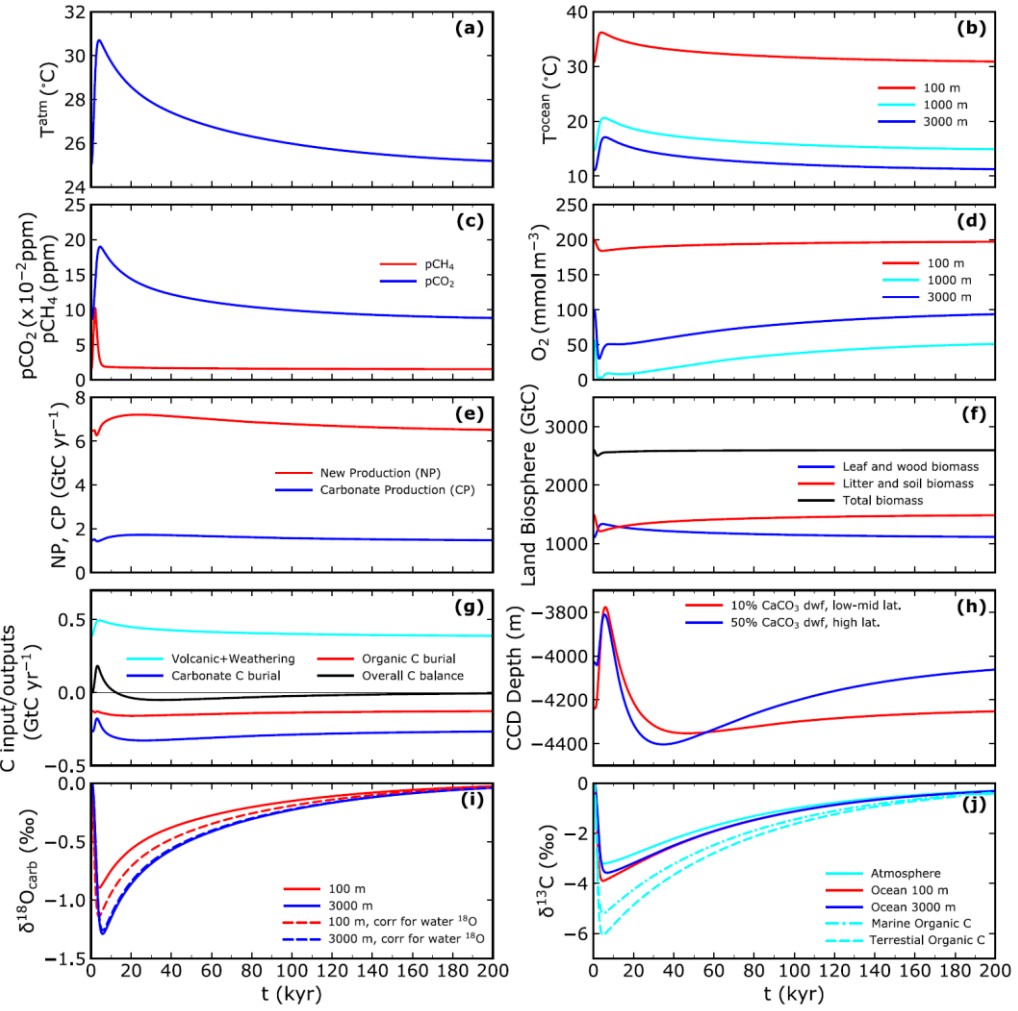

**Figure 4: Results from a 200,000 years simulation with a methane input of 4000 GtC over a time scale of 3000 years and with half the input dissolved in the ocean and half escaping as gas to the atmosphere.** **a**) Global mean atmospheric temperature, **b**) Low-mid latitude ocean temperature **c**) atmospheric $pCO_2$ and $pCH_4$, **d**) Low-mid latitude ocean dissolved $O_2$ concentration, **e**) Ocean new and biogenic $CaCO_3$ production, **f**) Land biosphere carbon inventories for Leaves+Wood, Litter+Soil and their total, **g**) External input/outputs of carbon. The black line is volcanic/weathering inputs minus burial outputs. **h**) Depths to the low-mid latitude 10 % (CCD) and high latitude 50 % $CaCO_3$ wt%, **i**) Low-mid latitude excursions of $\delta^{18}O$ in biogenic $CaCO_3$ formed as well as these excursions corrected for excursions in ambient $\delta^{18}O_w$. **j**) Carbon isotope excursions (CIE) for the atmosphere, the low-mid latitude ocean, low-mid latitude marine organic carbon and terrestrial organic carbon (see text; the methane input has $\delta^{13}C$ = -40 per mil).





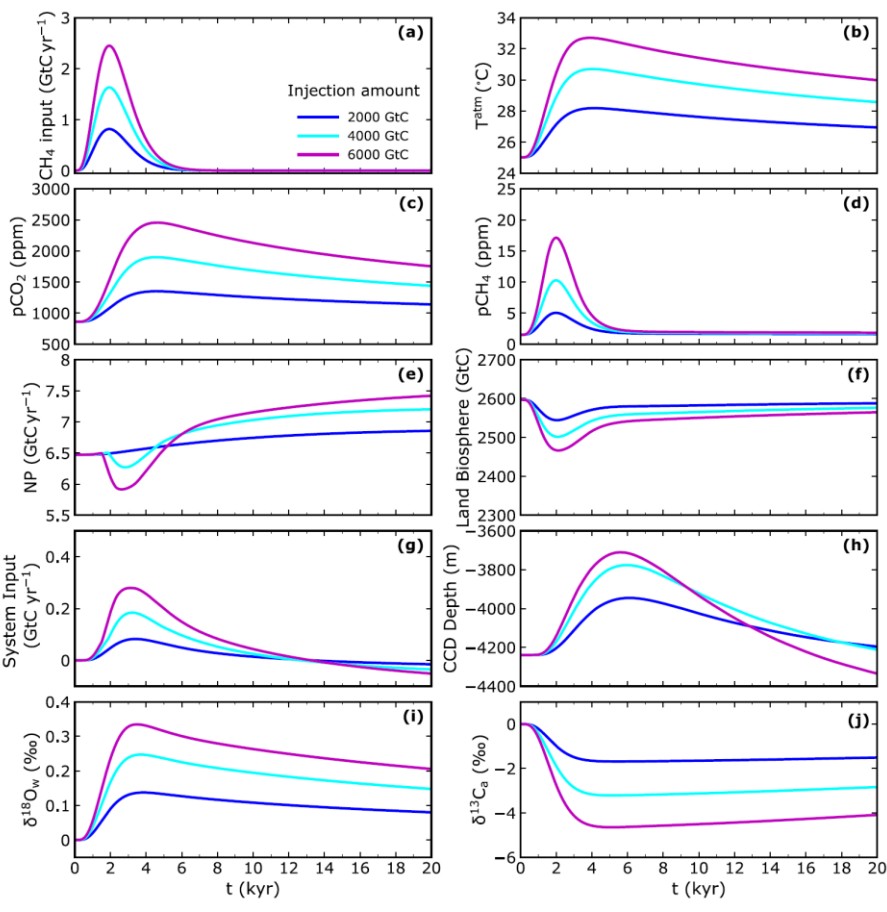

**Figure 5: Results for 20,000 year simulations for different methane input amounts over a time scale of 3000 years and with half the input dissolved in the ocean and half escaping as gas to the atmosphere. a**) methane input rate, **b**) global mean atmospheric temperature, **c**) atmospheric partial pressure of carbon dioxide, **d**) atmospheric partial pressure of methane, **e**) total ocean new production, **f**) total land biosphere biomass, **g**) volcanic/weathering inputs minus burial outputs (as black line in Fig. 4g), **h**) Low-mid latitude CCD depth (10% CaCO$_3$ dry weight in sediment), **i**) Oxygen isotope excursion of low-mid latitude, ocean surface layer water, **j**) Atmospheric carbon isotope excursion (the methane input has $\delta^{13}$C = -40 per mil).





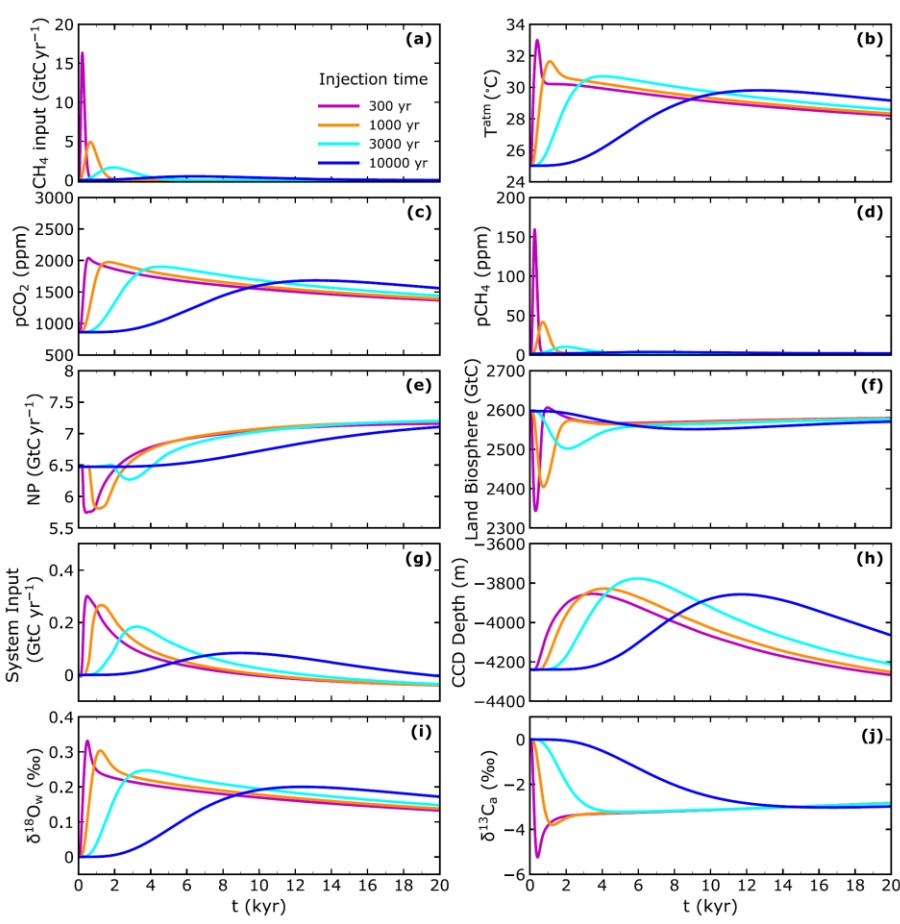

**Figure 6: Results for 20,000 year simulations for different methane input time scales for an input of 4000 GtC with half the input dissolved in the ocean and half escaping as gas to the atmosphere.** Properties plotted in **a**) – **j**) as in Fig. 5.



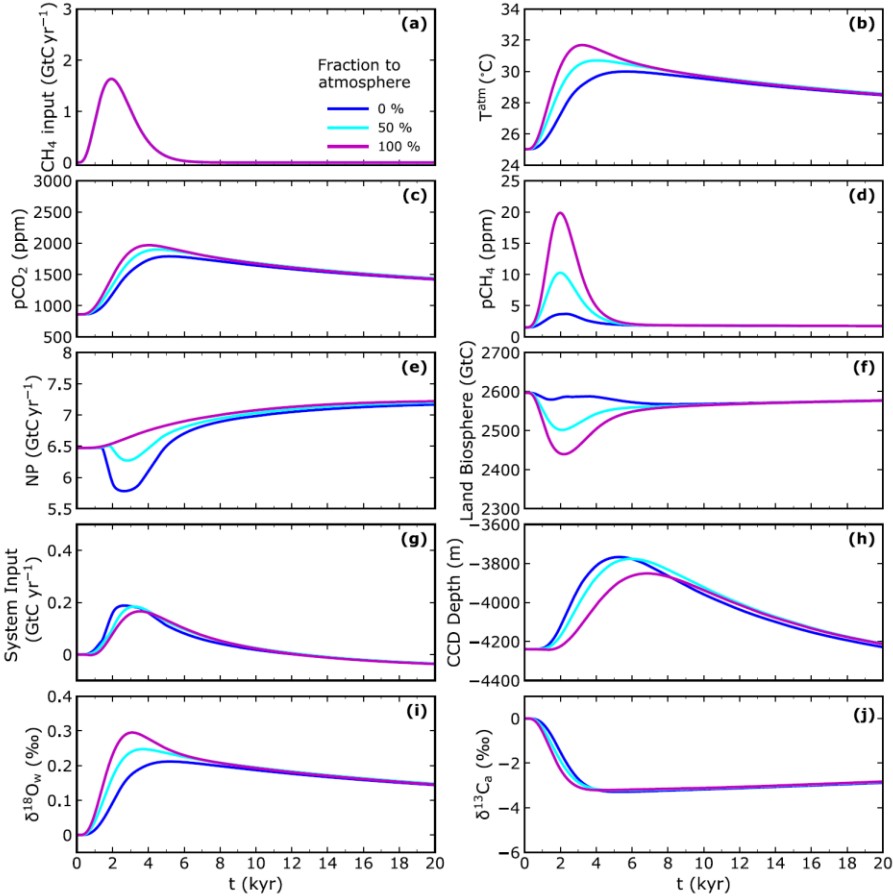

**Figure 7: Results for 20,000 year simulations for different fractions of methane escape to the atmosphere for a methane input of 4000 GtC over a time scale of 3000 years.** Properties plotted in **a**) – **j**) as in Fig. 5.





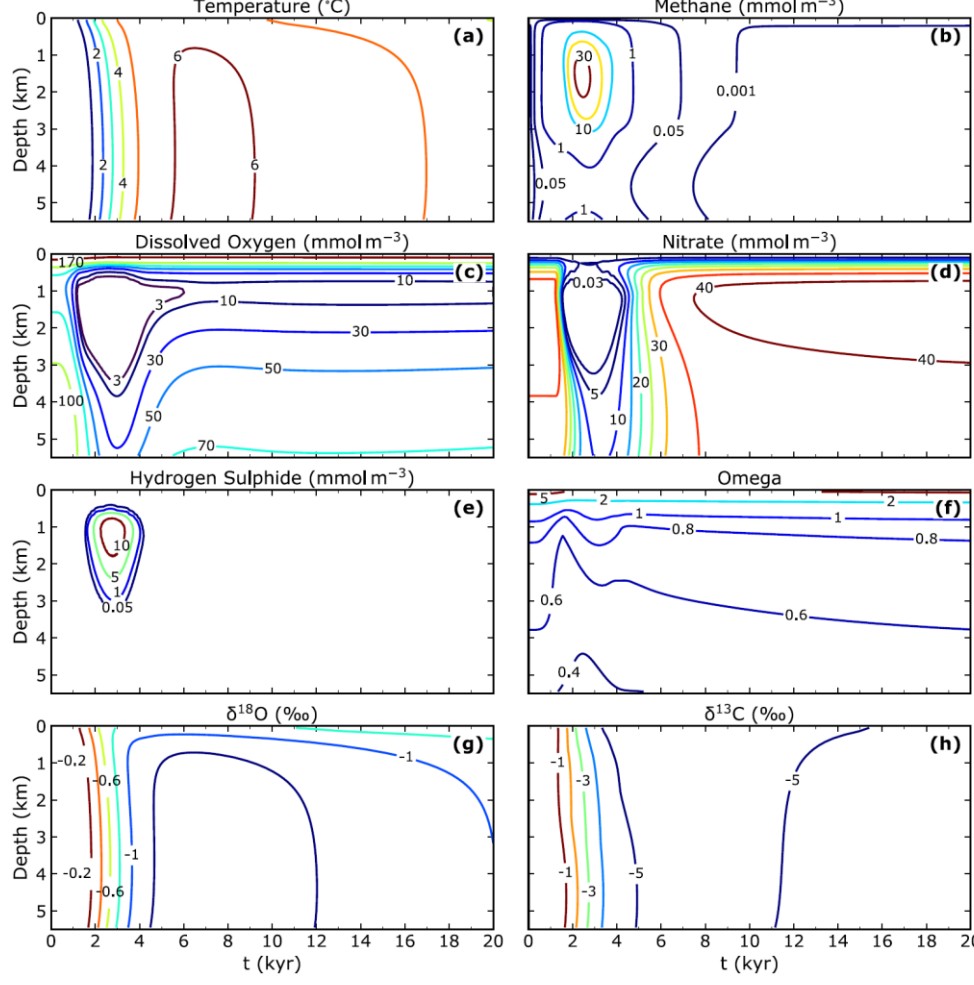

**Figure 8: Low-mid latitude ocean property results as functions of water depth and time for a 20,000 year simulation with a methane input of 6000 GtC over a time scale of 3000 years all of which dissolves in the ocean. a**) Temperature anomaly, **b**) methane concentration,, **c**) dissolved oxygen concentration (model denitrification for $O_2 < 3$ mmol m$^{-3}$), **d**) nitrate concentration (model sulphate reduction for $NO_3 < 0.03$ mmol m$^{-3}$), **e**) hydrogen sulphide concentration, **f**) Omega, the ratio of carbonate ion concentration to the carbonate ion saturation concentration for calcite (see Fig. 9g), **g**) Oxygen isotope excursion in biogenic carbonate produced in-situ. **h**) carbon isotope excursion. No corrections for the carbonate effect have been applied in **g** and **h** since these corrections were only derived for pelagic foraminifera (Spero et al., 1997),





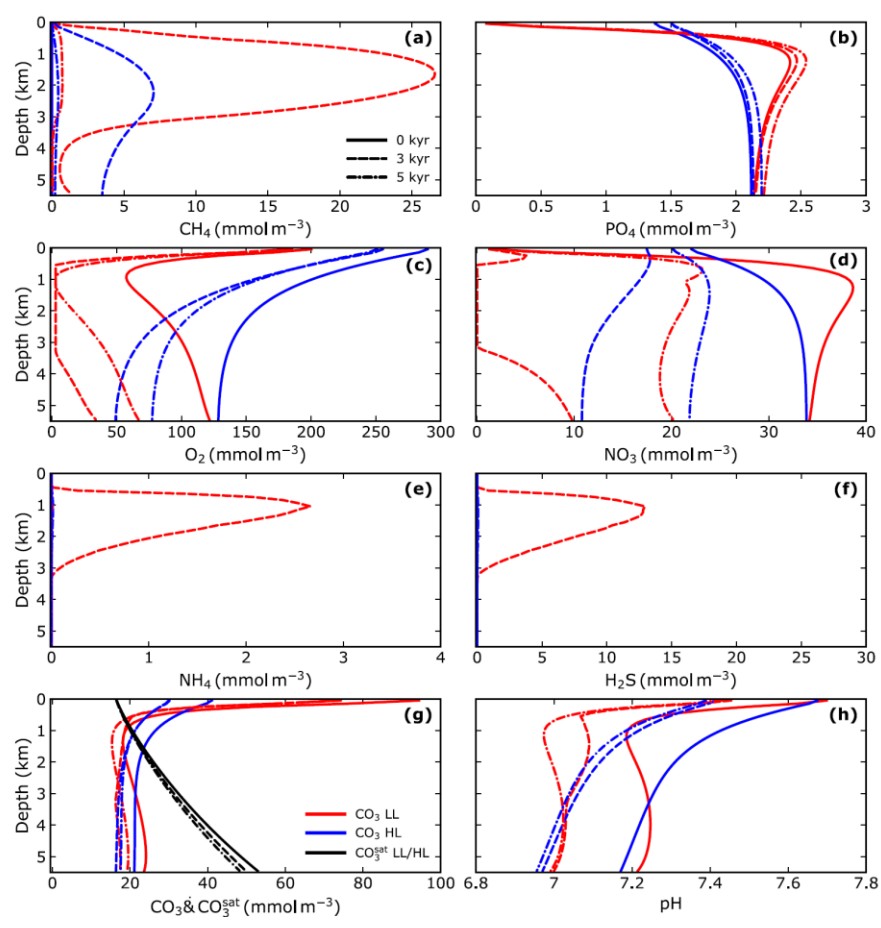

**Figure 9: Low-mid and high latitude ocean property profiles at selected times for the simulation of Figure 8.** Low-mid latitude (LL) and high latitude (HL) results are plotted with red and blue lines, respectively. Line types for the selected times after the simulation start (0, 3 and 5 kyr) are defined in pane **a**. $CO_3^{sat}$ is the carbonate ion saturation concentration for calcite.





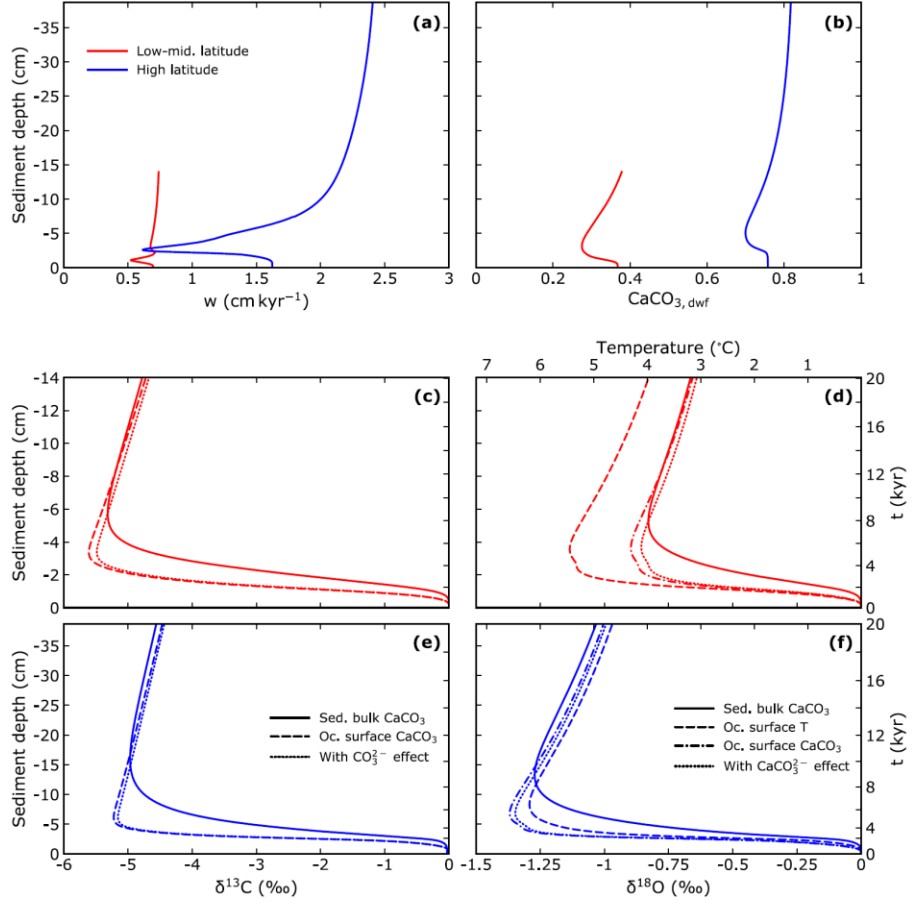

**Figure 10: Comparison of bulk carbonate results from model synthetic ocean sediment cores at 3000 m depth with ocean surface layer properties for the simulation of Figure 8.** Low-mid latitude (LL) and high latitude (HL) results are plotted with red and blue lines, respectively, and the methane input has $\delta^{13}$C = -40 per mil. **a**) burial (downward) velocity relative to the base of the bioturbated sediment layer vs. synthetic sediment depth (SSD). SSD is referenced to zero at the start of the simulation and increases downcore by convention, **b**) carbonate dry weight fraction vs. SSD, **c**) LL bulk carbonate $\delta^{13}$C excursion vs. LL SSD (solid line). Also shown are the $\delta^{13}$C excursion in biogenic carbonate in the LL ocean surface layer (dashed line) and this $\delta^{13}$C including the correction for change in surface layer $CO_3^{2-}$ (dashed-dot line; Spero et al., 1997), both plotted vs. simulation time (RHS y-axis). The time axis is "stretched/squeezed" relative to the SSD axis, a product of time-varying burial rates, **d**) LL bulk carbonate $\delta^{18}$O vs. LL SSD (solid line). Also shown are 1. ocean surface layer temperature anomaly (dashed line) on a temperature scale (top x-axis) related to the oxygen isotope scale (bottom x-axis) using temperature-dependence of biogenic carbonate $\delta^{18}$O (Bemis et al., 1998)), 2. $\delta^{18}$O in biogenic carbonate produced in the LL ocean surface layer (dashed-dot line) that includes the water $\delta^{18}$O excursion (see Figs 5-7 i) and 3. this $\delta^{18}$O including the correction for change in surface layer $CO_3^{2-}$ (dotted line; Spero et al., 1997), all plotted vs. simulation time, **e**) as **c** but for the HL ocean and sediment, **f**) as **d** but for the HL ocean and sediment.