# Peer review of "Implementation of methane cycling for deep time, global warming simulations with the DCESS Earth System Model (Version 1.2)"

_Geoscientific Model Development, 2017_

## Referee Comment (RC1) · D. Archer (Referee) · 3 Mar 2017

This is a thorough and polished description of the extension of an existing earth system climate model to deal with the impacts of massive methane releases on climate, ocean chemistry, and the carbon cycle. I don't think it's certain that methane is the culprit in all of the carbon isotope excursions in the past, but it might have been, and certainly a model of how this would work is a useful thing. Some of the pieces incorporated into the model are recent developments, like the IR absorption coefficients of overlapping gases. As this is new modeling territory, it is helpful to have a model description, in the spirit of GMD. The model description is generally detailed but some of the first-order information is omitted, like what an ocean sector is, or how the ocean circulates. I'm

sure it's described elsewhere but it should be here also. The results are clearly and attractively presented in the figures.

2. For me, encoding often-repeated phrases into acronyms like CIE, TM, and MH do not make a paper more readable, I don't see the point of it. It's easy to miss the definition, in which case you have to go back and find it, and even after you have it you have to train yourself to decode it every time.

17. Organic carbon could be liberated from a fossil organic source, or peat degradation; it need not be limited by the size of the terrestrial biosphere.

24. CO2 can also be released when magma intrudes into sediments. To the extent that it comes from CaCO3, it could tend to "dilute" the negative imprint of methane.

29. "sustain" solubility misleads, makes it sound like a process, rather than a concentration. Although it is a process, since there is an upward diffusive gradient, but this is not explained.

30. Clarify: the ocean model is 1-D (vertical)? What is an ocean sector? Explain the "high latitude zone" of the ocean (page 7). A diagram including the atmosphere and ocean would be very helpful.

4. It might be acknowledged here that much faster oxidation time scales are possible in places with ongoing methane availability, the biota builds up. Like the Gulf of Mexico.

---

## Referee Comment (RC2) · Anonymous Referee #2 · 5 Jul 2017

The authors have helped ground our theoretical understanding of past major CIEs by introducing methane cycling to an Earth System Model. I found the paper clear and thorough. My comments are few. The first is to reiterate David Archer's comments that some first order description of the model being modified would be useful – in the atmosphere, how is water vapor, heat transport, and temperature calculated? How does the ocean evolve? etc.

The second is that the relationship expressed in section 3.1 between the lifetime of atmospheric methane and its concentration seems to implicitly include an assumption of the water vapor concentration dependence on temperature, and a temperature

dependence on the methane concentration. Clarifying the nature of that assumption (what climate sensitivity is used?), and its robustness (does it matter the precise values chosen?), would be useful.

The third is that in the last paragraph of page 11, an exponential fit is used. It is not clear to me, but it seems possible that this fit is being applied in regions outside where the fit is performed, which seems like a potential issue.

The fourth is that I wish there was a more quantitative comparison of the changes induced by the new radiative forcing values, as well as a more explicit discussion of how the addition of the mechanisms that have been added to the Earth system model either confirm or complicate the more schematic picture paleoclimatologists may typically have of these events.

Finally, and building on the last point, following the causal chains of physical processes discussed in Chapter 4 can sometimes get confusing. It might be useful if there were some diagrams illustrating how the various graphed quantities from the figures influence (and feedback on) each other, illustrating the chain of connections leading out from the initial methane release. Comparing these diagrams between cases could then help clarify the qualitative difference between, for example, deep ocean and surface release of methane.

All in all this is a clear paper that shows how its modelling efforts have gained an important foothold on this problem, while also making clear what steps remain ahead. I endorse its publication.
* * *

---

## Author Comment (AC1) · 14 Jul 2017

First of all we would like to thank David Archer for reviewing our work and for his positive comments on it. Here follows our responses to specific points raised, points that are enclosed in quotes below.

". . . As this is new modeling territory, it is helpful to have a model description, in the spirit of GMD. The model description is generally detailed but some of the first-order information is omitted, like what an ocean sector is, or how the ocean circulates. I'm sure it's described elsewhere but it should be here also. . .".

The model is indeed described in great detail in the original DCESS publication (Shaffer et al, 2008, GMD). But in response to the comment above we now will also include several additional figures and additional text on our atmosphere and ocean modules in our revised version.

". . . .For me, encoding often-repeated phrases into acronyms like CIE, TM, and MH do not make a paper more readable, I don't see the point of it. It's easy to miss the definition, in which case you have to go back and find it, and even after you have it you have to train yourself to decode it every time. . ."

We can appreciate this point and in the revised version we will replace these acronyms and several additional ones by their full meanings in order to enhance readability.

". . . .Organic carbon could be liberated from a fossil organic source, or peat degradation; it need not be limited by the size of the terrestrial biosphere. . .".

In our model we treat weathering of fossil organic sources but this is a process too slow to contribute much carbon on the millennial time scales considered in our paper. Peat and permafrost would represent additional organic carbon sources. There would not be much if any permafrost carbon for the warm climate conditions we consider (see also discussion in Shaffer et al, 2016, GRL). Some additional carbon from peat degradation may be expected. We will discuss and quantify this in our revised version.

". . . .CO2 can also be released when magma intrudes into sediments. To the extent that it comes from CaCO3, it could tend to "dilute" the negative imprint of methane. .".

This is a good point and will be discussed in the revision, for example how the release of methane vs CO2 from organic carbon via magma intrusions may depend upon host rock composition or magma temperature. CO2 released from CaCO3 would put another wild card into the mix as will be discussed in the revision.

". . . . "sustain" solubility misleads, makes it sound like a process, rather than a concentration. Although it is a process, since there is an upward diffusive gradient, but this is

not explained. . ."

To clarify this, the text in the revised version will be changed to read ". . .Within a hydrate stability zone, methane hydrate is formed where methane release from bacterial remineralization of organic matter exceeds that needed to sustain solubility levels in the face of vertical diffusive transport of methane..".

".. Clarify: the ocean model is 1-D (vertical)? What is an ocean sector? Explain the "high latitude zone" of the ocean (page 7). A diagram including the atmosphere and ocean would be very helpful. . ."

See comment above about including additional figures and text in the revised version.

". . ..It might be acknowledged here that much faster oxidation time scales are possible in places with ongoing methane availability, the biota builds up. Like the Gulf of Mexico..."

This is also a good point and will be discussed in the revised version.

---

## Author Comment (AC2) · 14 Jul 2017

First of all we would like to thank Referee 2 for reviewing our work and for her/his positive comments on it. Here follows our responses to specific points raised, points that are enclosed in quotes below.

"...The first is to reiterate David Archer's comments that some first order description of the model being modified would be useful – in the atmosphere, how is water vapor, heat transport, and temperature calculated? How does the ocean evolve? etc....."

In response to similar comments by both referees, we now will also include several

additional figures and additional text on our atmosphere and ocean modules in our revised version.

"…..The second is that the relationship expressed in section 3.1 between the lifetime of atmospheric methane and its concentration seems to implicitly include an assumption of the water vapor concentration dependence on temperature, and a temperature dependence on the methane concentration. Clarifying the nature of that assumption (what climate sensitivity is used?), and its robustness (does it matter the precise values chosen?), would be useful…"

As described in our manuscript the relationship we derive between the lifetime of atmospheric methane and its concentration is based on fitting a simple generic function (Shaffer et al, 2008, GMD) to results from three different studies using models that include complex atmospheric chemistry (Schmidt and Shindell 2003, Paleoceanography; Lamarque et al 2006, Paleoceanography; Isaksen et al, 2011, Global Biogeochemical Cycles). The results of these models imply that the link "water vapor concentration dependence on temperature, and a temperature dependence on the methane concentration" does not affect their (nor our) results significantly. For example, Schmidt and Shindell write "In the Paleocene, warmer temperatures will have likely lead to increased atmospheric water amounts that affect the production of OH. An estimate of H2O increases in a greenhouse world would be about 30% for a 4°C temperature increase, making the reasonable assumption that relative humidity is roughly constant [IPCC, 2001]. Over a range of GCM experiments, the increase in OH for this magnitude change is less than 10% [Grenfell et al.,2001, personal communication]. This actually leads to an increased sensitivity to CH4 changes, though not by a significant amount." And Isaksen et al. write "In the calculations we have used current atmospheric water vapor content. Since water vapor is expected to increase in a future warmer climate the calculations were repeated for a 40% increase in tropospheric water vapor (but no other changes). ….. We found that the calculated tracer and lifetime perturbations were only slightly affected by this increase (less than 10% impact)". We will mention

this insensitivity in our revised version.

".…The third is that in the last paragraph of page 11, an exponential fit is used. It is not clear to me, but it seems possible that this fit is being applied in regions outside where the fit is performed, which seems like a potential issue…"

The atmospheric temperature results we report in our manuscript all lie within the range 0 to 40°C. The exponential fit we derive to data spanning this range with 5°C increments is excellent ($R^2$ = 0.997) so we see no issue here.

".…The fourth is that I wish there was a more quantitative comparison of the changes induced by the new radiative forcing values, as well as a more explicit discussion of how the addition of the mechanisms that have been added to the Earth system model either confirm or complicate the more schematic picture paleoclimatologists may typically have of these events".

In response to the first part of this comment, in our revision we will extend the discussion found in the first full paragraph on page 6 to compare warming that would be calculated using the old and new radiative forcing values for several specific combinations of $pCO_2$ and $pCH_4$. With regard to the second part of the comment, this is a good idea. However we feel that this is better covered in the follow-up manuscripts on specific deep-time, global warming events that we're working on rather than in the present model description paper.

".…Finally, and building on the last point, following the causal chains of physical processes discussed in Chapter 4 can sometimes get confusing. It might be useful if there were some diagrams illustrating how the various graphed quantities from the figures influence (and feedback on) each other, illustrating the chain of connections leading out from the initial methane release. Comparing these diagrams between cases could then help clarify the qualitative difference between, for example, deep ocean and surface release of methane...."

The figures in Chapter 4 are our best attempt, after much thought and effort, to present these types of relationships in a clear, comprehensive and attractive way; referee 1 thinks that we have succeeded with this ("...The results are clearly and attractively presented in the figures.."). We feel that figures along the lines requested would fit in better in our work in progress outlined above.

————————————————————

**[GMDD](GMDD)**

---

## Author Response (AR1)

**Response to reviewers' comments on and revision of our manuscript "*Implementation of methane cycling for deep time, global warming simulations with the DCESS Earth System Model (Version 1.2)*"**

5 In the following we respond to the comments of both reviewers (in red) and indicate revisions we made in our original manuscript to address these comments At the end of this file is a markup copy of our manuscript. . Line and page numbers below refer to this markup copy.

**Response to Reviewer 1 (David Archer)**

1. "… As this is new modeling territory, it is helpful to have a model description, in the spirit of GMD. The model description is generally detailed but some of the first-order information is omitted, like what an ocean sector is, or how the ocean circulates. I'm sure it's described elsewhere but it should be here also…".

The model is indeed described in great detail in the original DCESS publication (Shaffer et al, 2008, GMD). But in response to the comment above we now will also include several additional figures and additional text on our atmosphere and ocean modules in our revised version. In particular we now include new figures 2 and 3 that show the basic model geometry (including what an ocean sector is) and

20 the atmosphere, ocean and ocean sediment modules (including how the ocean mixes and circulates. Furthermore we have expanded of model description text to provide more in-depth, first order information on the model (from line 25, page 3 to line 4 of page 6).

2. "….For me, encoding often-repeated phrases into acronyms like CIE, TM, and MH do not make a

25 paper more readable, I don't see the point of it. It's easy to miss the definition, in which case you have to go back and find it, and even after you have it you have to train yourself to decode it every time…"

We can appreciate this point and in the revised version we replaced the acronyms CIE, TM and MH throughout with their full meanings. Furthermore, we also replaced some other acronyms like IPCC, PI, and POM by their full meanings where we found that to be appropriate.

3. "....Organic carbon could be liberated from a fossil organic source, or peat degradation; it need not be limited by the size of the terrestrial biosphere…".

In our model we treat weathering of fossil organic sources but this is a process too slow to contribute much carbon on the millennial time scales considered in our paper. Peat and permafrost would represent additional organic carbon sources. There would not be much if any permafrost carbon for the warm climate conditions we consider but some additional carbon from peat degradation might be expected. We discussed and quantified the above in our revised version (including several new references) on lines 20-24 of page 2 that now read "…Permafrost and peat carbon are other possible sources of terrestrial carbon. Little permafrost could exist for the warm conditions leading up to the warming events considered here (Shaffer et al, 2016). An explanation in terms of global peat conflagration (Kurtz et al., 2003) or degradation seems unlikely since it would require peat carbon reserves almost 20 times greater than present reserves of around 500 GtC (Brigham et al., 2006).". Thus we find significant peat and permafrost sources to be unlikely.

4. "….CO2 can also be released when magma intrudes into sediments. To the extent that it comes from CaCO3, it could tend to "dilute" the negative imprint of methane..".

This is a good point and we discussed it in the revision (including new references) in lines 28-33, page 2 that now read: "….Thermogenic methane and carbon dioxide are produced when magma intrudes into overlying sediments containing old organic carbon. More such intrusions would be expected in association with increased volcanism that creates large igneous provinces and such provinces tend to correlate in time with the ancient carbon isotope excursions (Svensen et al., 2004). Magma contact with shale and coal favor production of $CH_4$ and $CO_2$, respectively, whereas such contact with (isotopicallyheavy) limestone produces very limited amounts of $CH_4$ and $CO_2$ (Arnes et al., 2011). Shale is typically the main host rock through which magma intrudes in large igneous provinces (Svensen et al. 2004; Svensen et al., 2007)."

5 From this it can be concluded that $CH_2$ was likely the main gas released and that little $CH_4$ or $CO_2$ was likely emitted from magma contact with $CaCO_3$.

5. "….. "sustain" solubility misleads, makes it sound like a process, rather than a concentration. Although it is a process, since there is an upward diffusive gradient, but this is not explained…"

To clarify this, the revised the text to read (lines 2-3, page 3): ".. Within a hydrate stability zone, methane hydrate is formed when methane release from bacterial remineralization of organic matter exceeds that needed to sustain solubility levels in the face of vertical diffusive transport of methane…"

15 6. ".. Clarify: the ocean model is 1-D (vertical)? What is an ocean sector? Explain the "high latitude zone" of the ocean (page 7). A diagram including the atmosphere and ocean would be very helpful…

As mentioned above under point 1 we have now included two new figures as well as additional text that
20 fully address these concerns.

7. "….It might be acknowledged here that much faster oxidation time scales are possible in places with ongoing methane availability, the biota builds up. Like the Gulf of Mexico..."

25 This comment prompted us to test the sensitivity of our model to much fast methane oxidation times. First we inserted an additional text (with a new reference) in lines 7-10, page 11 that read "…A much shorter lifetime of ∼ 4 months was deduced for strong, fast and localized methane emissions during the 2010 Deepwater Horizon oil spill in the Gulf of Mexico (Kessler et al., 2011). Below we take $\tau_{ox,CH_4}$ to

be 50 years since this value is likely more applicable to our global approach. But in the following we also check the sensitivity of our results to the much shorter lifetime".

Second, we carried out an additional simulation using this much faster oxidation time, plotted the results in new Fig. 9 and discussed these new results in lines 19-26, Page 18 that read "We also carried out a simulation like the case in Fig. 9 for all methane dissolving in the ocean (blue lines) but for a much shorter oceanic methane lifetime of 4 months (compared to the standard lifetime of 50 years; see Section 3.2.2). Results for this new simulation are plotted as dotted blue lines in Fig. 9. The shorter lifetime leads to much reduced methane concentrations in the ocean and less outgassing to the atmosphere (Fig. 9d). Furthermore, faster methane oxidation in the ocean decreases dissolved oxygen there slightly, leading to slightly enhanced anoxia, sulphate-dependent, anoxic methane oxidation and alkalinity inputs. As a consequence, CCD shoaling is damped somewhat as is the atmospheric $p$CO2 increase (Fig. 9c,h). Together with less methane outgassing, this leads to a reduction of maximum global warming of about 7% compared to the standard case (Fig. 9b). ".

**Response to Reviewer 2**

8. "…The first is to reiterate David Archer's comments that some first order description of the model being modified would be useful – in the atmosphere, how is water vapor, heat transport, and temperature calculated? How does the ocean evolve? etc….."

As mentioned above under points 1 and 6 above we have now included two new figures as well as additional text that address these concerns.

9. "….The second is that the relationship expressed in section 3.1 between the lifetime of atmospheric methane and its concentration seems to implicitly include an assumption of the water vapor concentration dependence on temperature, and a temperature dependence on the methane concentration.

Clarifying the nature of that assumption (what climate sensitivity is used?), and its robustness (does it matter the precise values chosen?), would be useful…"

As described in our manuscript the relationship we derive between the lifetime of atmospheric methane and its concentration is based on fitting a simple generic function (Shaffer et al, 2008, GMD) to results from three different studies using models that include complex atmospheric chemistry (Schmidt and Shindell 2003, Paleoceanography; Lamarque et al 2006, Paleoceanography; Isaksen et al, 2011, Global Biogeochemical Cycles). The results of these models imply that the link "water vapor concentration dependence on temperature, and a temperature dependence on the methane concentration" does not affect their (nor our) results significantly. For example, Schmidt and Shindell write "In the Paleocene, warmer temperatures will have likely lead to increased atmospheric water amounts that affect the production of OH. An estimate of $H_2O$ increases in a greenhouse world would be about 30% for a 4°C temperature increase, making the reasonable assumption that relative humidity is roughly constant [IPCC, 2001]. Over a range of GCM experiments, the increase in OH for this magnitude change is less than 10% [Grenfell et al.,2001, personal communication]. This actually leads to an increased sensitivity to $CH_4$ changes, though not by a significant amount." And Isaksen et al. write "In the calculations we have used current atmospheric water vapor content.  Since water vapor is expected to increase in a future warmer climate the calculations were repeated for a 40% increase in tropospheric water vapor (but no other changes). ….. We found that the calculated tracer and lifetime perturbations were only slightly affected by this increase (less than 10% impact)".

In our revision we discuss this point in lines 21-29, page 6 that read "… For the range of low to medium methane concentrations considered in Schmidt and Shindell (2003) and Isaksen et al. (2011), these authors found rather low sensitivity of their results to atmospheric water vapor levels citing methane lifetime changes of less than 10% for tropospheric water vapor increases of 30 − 40%. On the other hand, Lamarque et al., (2006) found that atmospheric lifetimes decrease for extreme values of $p$CH$_4$ due to much enhanced water vapor and OH production (and thereby greater methane oxidation) of a very warm climate. The red line in Fig. 2 shows a fit to all the chemistry modeling results of the figure using

Eq. 1, a fit for which $\tau_{PI} = 9.5$ years, $a = 0.78$ and $b = 11$. This new fit now also captures in part enhanced OH production and associated limitation of $\tau$ for very high values of $pCH_4$ and has been adopted in our enhanced model whereby the model atmospheric methane sink in moles per year is $\upsilon_a pCH_4 / \tau$ with $\upsilon_a$ as the atmospheric mole content...".

10. "....The third is that in the last paragraph of page 11, an exponential fit is used. It is not clear to me, but it seems possible that this fit is being applied in regions outside where the fit is performed, which seems like a potential issue..."

The atmospheric temperature results we report in our manuscript all lie within the range 0 to 40°C. The exponential fit we derive to data spanning this range with 5°C increments is excellent ($R^2 = 0.997$) so we see no issue here and we took no further action.

11. "....The fourth is that I wish there was a more quantitative comparison of the changes induced by the new radiative forcing values, as well as a more explicit discussion of how the addition of the mechanisms that have been added to the Earth system model either confirm or complicate the more schematic picture paleoclimatologists may typically have of these events".

In response to the first part of this comment, in our revision we extended the discussion from our original manuscript on this to compare warming that would be calculated using the old and new radiative forcing values for several specific combinations of $pCO_2$ and $pCH_4$. Thus in the revised manuscript, lines 11-24, page 7 now read "...To illustrate the effects of these new radiative forcing formulations we consider several specific atmosphere compositions and warming associated with them. If we consider an atmosphere with $pCO_2$, $pCH_4$ and $pN_2O$ values of 500, 10 and 0.5 ppm, respectively, this corresponds to radiative forcings relative to pre-industrial conditions of 3.26, 2.32 and 0.73 W/m$^2$ but with a total forcing overlap of -0.16 W/m$^2$, leading to a total forcing of 6.15 W/m$^2$. With the original radiative forcing formulations used in the DCESS model (Myhre et al., 1998; Shaffer et al., 2008) the total radiative forcing is similar, 6.25 W/m$^2$. For a nominal climate sensitivity of 3°C for a $pCO_2$

doubling (0.81°C/W/m$^2$), such total forcings would lead to global mean temperature increases of 4.98 and 5.06 °C, respectively, above the DCESS pre-industrial mean global temperature of 15°C. If we consider an atmosphere with $p$CO$_2$, $p$CH$_4$ and $p$N$_2$O values of 1000, 100 and 1 ppm, respectively, this corresponds to radiative forcings relative to pre-industrial conditions of 7.45, 6.50  and 1.88 W/m$^2$ but

5    with a total forcing overlap of -0.93 W/m$^2$, leading to a total forcing of 14.97 W/m$^2$. With the original radiative forcing formulations, the total radiative forcing is considerably higher, 18.36 W/m$^2$. For a 3°C climate sensitivity, such total forcing would lead to global mean temperature increases of 12.13 and 14.87 °C, respectively; for a 5°C climate sensitivity (Shaffer et al, 2016) the increases would be 20.22 and 24.79 °C, respectively…"

With regard to the second part of comment 11, we find this to be a good idea but we feel that this is better covered in the follow-up manuscripts on specific deep-time, global warming events that we're working on rather than in the present model description paper.

15    12. "….Finally, and building on the last point, following the causal chains of physical processes discussed in Chapter 4 can sometimes get confusing. It might be useful if there were some diagrams illustrating how the various graphed quantities from the figures influence (and feedback on) each other, illustrating the chain of connections leading out from the initial methane release.  Comparing these diagrams between cases could then help clarify the qualitative difference between, for example, deep

20    ocean and surface release of methane...."

The figures in Chapter 4 are our best attempt, after much thought and effort, to present these types of relationships in a clear, comprehensive and attractive way; referee 1 thinks that we have succeeded with this ("…The results are clearly and attractively presented in the figures.."). We feel that further figures

25    along the lines requested in comment 12 would fit in better in our work in progress outlined above and thus we have taken no further action here.

[revised manuscript text omitted]

---

## Author Response (AR2)

Technical correction to manuscript " **Implementation of methane cycling for deep time, global warming simulations with the DCESS Earth System Model (Version 1.2)"**

In response to the suggestion by the Topical Editor, we made an included a new schematical figure (new Figure 10) that illustrates differences in pathways and feedbacks for methane input to the atmosphere versus methane input to the ocean. A short text referring to the new figure has been added to the end of the next-to-last paragraph in Section 4.3 of the paper and reads "…..The flow diagram presented in Fig. 10 serves to compare and contrast the pathways and feedbacks associated with 1) methane input to the ocean only and 2) methane input to the atmosphere only. These are the cases plotted as blue and maroon lines, respectively, in Fig. 9."